# SUMOylation of Na$_V$1.2 channels mediates the early response to acute hypoxia in central neurons

Leigh D Plant[1], Jeremy D Marks[2], Steve AN Goldstein[1]*

[1]Department of Biochemistry, Brandeis University, Waltham, United States;
[2]Department of Pediatrics, University of Chicago, Chicago, United States

**Abstract** The mechanism for the earliest response of central neurons to hypoxia—an increase in voltage-gated sodium current ($I_{Na}$)—has been unknown. Here, we show that hypoxia activates the Small Ubiquitin-like Modifier (SUMO) pathway in rat cerebellar granule neurons (CGN) and that SUMOylation of Na$_V$1.2 channels increases $I_{Na}$. The time-course for SUMOylation of single Na$_V$1.2 channels at the cell surface and changes in $I_{Na}$ coincide, and both are prevented by mutation of Na$_V$1.2-Lys38 or application of a deSUMOylating enzyme. Within 40 s, hypoxia-induced linkage of SUMO1 to the channels is complete, shifting the voltage-dependence of channel activation so that depolarizing steps evoke larger sodium currents. Given the recognized role of $I_{Na}$ in hypoxic brain damage, the SUMO pathway and Na$_V$1.2 are identified as potential targets for neuroprotective interventions.

## Introduction

Acute hypoxia contributes to brain damage arising from such common conditions as stroke, heart attack, and head trauma. In humans, decreased cerebral blood flow (ischemia) resulting in acute neuronal hypoxia correlates with pathological changes in electroencephalographic recordings and decreased electrical signaling in less than 150 s (*Sundt et al., 1981*). At the cellular level, the first effect of hypoxia is to increase $I_{Na}$ (*Boening et al., 1989*; *Stys et al., 1992*); this precedes a series of events that include depolarization of the plasma membrane, excitotoxic elevation of intracellular calcium, mitochondrial dysfunction, ATP depletion, increased production of reactive oxygen species and, ultimately, cell death (*Leao, 1944*; *Hansen, 1985*; *Choi, 1990*). While these downstream effects have been well studied, the early hypoxia-induced change in Na$^+$ flux has received less attention despite strong evidence to support its critical role in the hypoxic insult: inhibition of $I_{Na}$ by tetrodotoxin (TTX) attenuates hypoxia-induced depolarization and reduces neuronal death in the hippocampus, hypothalamus, and neocortex (*Boening et al., 1989*; *Stys et al., 1992*; *Weber and Taylor, 1994*; *Xie et al., 1994*; *Taylor et al., 1995*; *Fung et al., 1999*; *Horn and Waldrop, 2000*; *Raley-Susman et al., 2001*; *Banasiak et al., 2004*). Furthermore, the neuroprotective effects of TTX have been judged to occur both independent of, and by reduction of the excitotoxic effects that follow $I_{Na}$-induced membrane depolarization.

In excitable cells, membrane depolarization opens voltage-gated sodium (Na$_V$) channels, initiating an explosive influx of sodium ions (Na$^+$) that generate the rising phase of the action potential (*Catterall, 2000*). These channels are mixed complexes comprised of one α-subunit (~2000 residues), which contains four voltage sensor domains and one ion conduction pore, and smaller β-subunits that modify function. Of the ten genes for α-subunits in mammals, four are predominant in the central nervous system including SCN2a, which encodes Na$_V$1.2, an α-subunit that is widely distributed in the brain. SCN2a mutations are associated with epilepsy and febrile seizures (*Shi et al., 2012*).

*For correspondence: goldstein@brandeis.edu

**Competing interests:** The authors declare that no competing interests exist.

**eLife digest** Neurons in the brain require a continuous supply of oxygen for normal activity. If the level of oxygen in the brain decreases – for example when a blood vessel becomes blocked – neurons begin to die, and permanent brain damage can result. A shortage of oxygen first causes sodium ion channels within the surface membrane of the neurons to open. Sodium ions then flow into the cells through these open channels to trigger a cascade of events inside the cells that ultimately results in their death.

Plant et al. now reveal how oxygen deficiency, otherwise known as hypoxia, rapidly increases the flow of sodium ions into brain cells. By inducing hypoxia in neurons from the rat brain, Plant et al. show that a lack of oxygen causes the SUMOylation – a process whereby a series of enzymes work together to attach a Small Ubiquitin-like Modifier (or SUMO) protein – of specific sodium ion channels in under a minute. The channels linked to the SUMO protein, a subtype called $Na_V1.2$, open more readily than unmodified channels, allowing more sodium ions to enter the neurons.

Plant et al. study granule cells of the cerebellum, the most numerous type of neuron in the human brain. Further investigation is required to determine if SUMOylation of $Na_V1.2$ channels underlies the response of other neurons to hypoxia as well. It also remains to be discovered whether molecules that block the SUMOylation of $Na_V1.2$ channels, or that prevent the flow of sodium ions through these channels, could reduce the number of brain cells that die in low-oxygen conditions such as strokes.

SUMOylation is the enzyme-mediated, post-translational linkage of one of three SUMO isoforms to the ε-amino group of specific Lys residues on a target protein (*Henley et al., 2014*). Present in all eukaryotic cells, the SUMO pathway was recognized to regulate the trafficking and activity of nuclear transcription factors when we discovered it to operate as well at the plasma membrane to regulate neuronal excitability via direct SUMOylation of $K^+$ channel α-subunits (*Rajan et al., 2005*; *Plant et al., 2010*, *2011*, *2012*). Here, seeking to determine if the SUMO pathway also regulated sodium channels, we documented SUMOylation of $Na_V1.2$ channels in CGN and then recognized it to be the basis for acute, hypoxic augmentation of $I_{Na}$.

To establish the mechanism, we studied rat CGN using whole-cell voltage-clamp, and observed hypoxia to increase $I_{Na}$ to a new steady-state level in <40 s due to an excitatory, leftward shift in the voltage required to activate the current. The shift was recapitulated by application of SUMO1 under normoxic conditions and suppressed by the deSUMOylating enzyme SENP1. Consistent with tonic control of $I_{Na}$ by the SUMO pathway, SUMO1 and SENP1 increased and decreased the current, respectively, under normoxic conditions. The response of $I_{Na}$ to hypoxia was ablated by μ–Cono-toxin-TIIIA (CnTX), a potent blocker of $Na_V1.2$ channels. Supporting the implied mechanism—rapid SUMOylation of $Na_V1.2$ channels at the CGN plasma membrane in response to hypoxia—hypoxia was directly shown to increase the interaction of native SUMO1 and $Na_V1.2$ at the neuronal surface using antibody-mediated fluorescent resonance energy transfer (amFRET) microscopy and ground state depletion, stochastic optical reconstruction super-resolution microscopy (STORM). SUMOylation of $Na_V1.2$ on Lys38 was shown to be necessary and sufficient to explain the changes in $I_{Na}$ induced by hypoxia by reconstitution of the hypoxic response in Chinese Hamster Ovary (CHO) cells using heterologously expressed subunits. Further, study of live CHO cells in real-time using total internal reflection fluorescence (TIRF) microscopy revealed that acute hypoxia leads to monoSUMOylation of single $Na_V1.2$ channels already at the plasma membrane without a change in the number of channels on the surface.

## Results

### Hypoxia rapidly increases CGN $I_{Na}$

In ambient (21%) $O_2$, CGN $I_{Na}$ activated and inactivated rapidly, showing the expected biophysical properties (*Diwakar et al., 2009*), including a mean peak $I_{Na}$ of $-172 \pm 20$ pA/pF at $-20$ mV, a half-maximal activation voltage ($V_{1/2}$) of $-23 \pm 0.5$ mV, and a steady-state inactivation midpoint (SSI) of

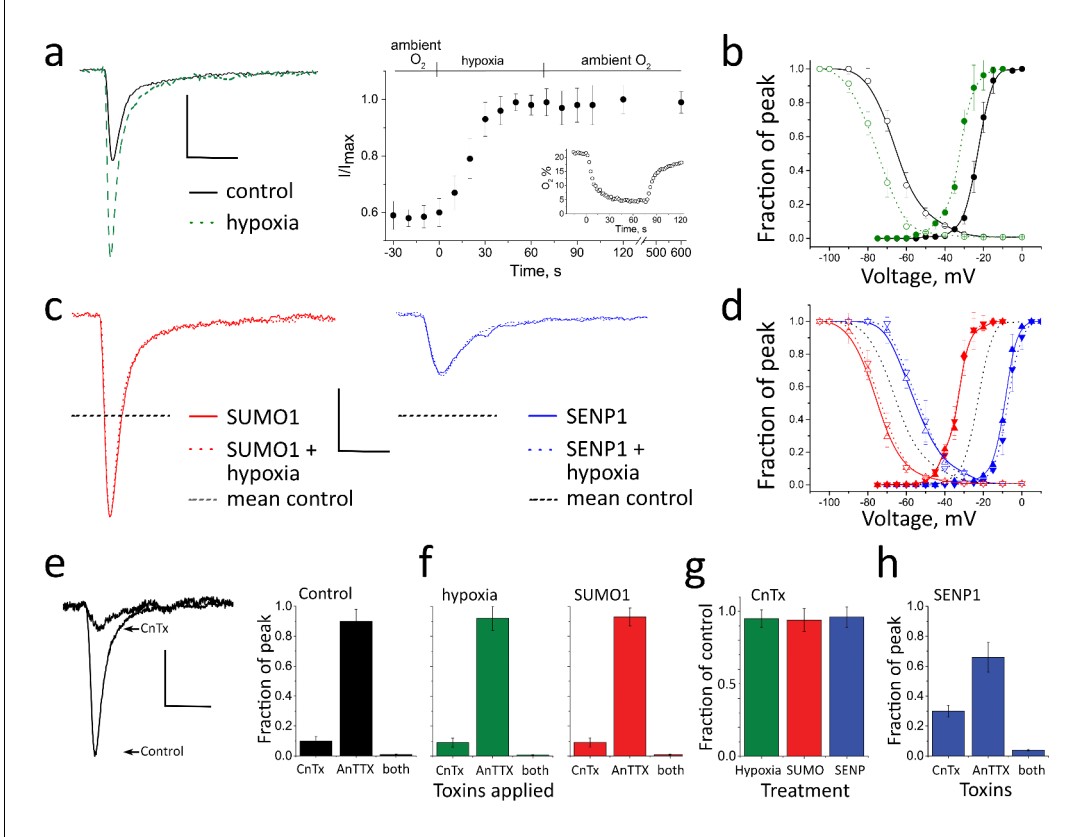

**Figure 1.** Acute hypoxia and SUMO1 augment $I_{Na}$ in rat CGN. $I_{Na}$ in rat CGN was studied by whole-cell patch-clamp. Normalized activation (Act) and steady-state inactivation (SSI) relationships were obtained and fit as described in the Materials and methods. Measured values are noted in the text and listed in **Table 1**. The time-course of hypoxic modulation of $I_{Na}$ was studied by steps from –100 mV to -20 mV every 10 s. Cells were studied with control solution (black), 100 pm SUMO1 (red), or 250 pm SENP1 (blue) in the recording pipette. To assess the relative contributions of $Na_V1.2$ and $Na_V1.6$ channel currents to $I_{Na}$, 250 nm μ-conotoxin TIIIA (CnTx) and 50 nm 4,9 anhydro-TTX (anTTX) were applied as indicated. Data are mean ± S.E.M. for 10 to 15 cells per group. Scale bars are 150 pA/pF and 5 ms for panels **a** to **d**, and 75 pA/pF and 5 ms in panel **e**. (**a**) Left, example traces showing $I_{Na}$ at -20 mV increased when control perfusate at 21% $O_2$ (black) was exchanged with a hypoxic solution at 5% $O_2$ (- - - -). Right, the time-course for changes in the peak current in response to decreased $O_2$, and on return to normoxia (washout), is shown normalized to the maximal current for each cell studied. Inset, The level of $O_2$ was measured in real-time in the recording chamber and fell from 21 to 5% within 18 ± 1 s (o). Error bars are within the symbols. (**b**) Hypoxic solution (green dashes) left shifted the $V_{1/2}$ of $I_{Na}$ from control (black) for both Act (solid) and SSI (open). (**c**) Left, SUMO1 in the pipette (red) increased $I_{Na}$ and no further augmentation was observed by subsequent hypoxia (red dashes). Right, SENP1 in the pipette (blue) decreased $I_{Na}$ and suppressed the response to hypoxia (blue dashes). Dotted black lines indicate the mean peak values in control normoxic solutions from **a**. (**d**) SUMO1 left shifted the $V_{1/2}$ of Act (red triangle) and SSI (open red triangle). The relationships were then insensitive to hypoxia (Act, red down triangle; SSI, open red down triangle). SENP1 right shifted the $V_{1/2}$ of Act (blue triangle) and SSI (open blue triangle) and currents were then insensitive to hypoxia. Dashed black lines indicate mean peak values with control solutions from **b**. (**e**) Left; $I_{Na}$ traces under control conditions and with toxins in the bath. Right; mean normalized peak current histograms showing 90 ± 0.8% inhibition by CnTx, 10 ± 0.3% inhibition by AnTTX and 98 ± 0.02% inhibition by both toxins with 21% $O_2$ (black bars). (**f**) Mean normalized peak $I_{Na}$ histograms. Left; 92 ± 1% inhibition by CnTx and 9 ± 1% inhibition by AnTTX with a drop to 5% $O_2$ for 60 s (green). Right; 93 ± 2% inhibition by CnTx and 7 ± 1% inhibition by AnTTX with 100 pm SUMO1 in the pipette (red). (**g**) Mean normalized peak current histograms show that the $I_{Na}$ remaining after inhibition of $Na_V1.2$ by CnTx did not respond to acute hypoxia at 5% $O_2$ (green), SUMO1 (red) or SENP1 (blue). (**h**) Mean normalized peak $I_{Na}$ histograms with SENP1 in the pipette with 21% $O_2$ (blue) showing 70 ± 4% inhibition by CnTx, 34 ± 8% inhibition by AnTTX and 96 ± 0.5% inhibition by both toxins consistent with the passage of much of the remaining current by $Na_V1.6$.

The following figure supplement is available for figure 1:

**Figure supplement 1.** Recovery of $I_{Na}$ and $Na_V1.2$ from fast inactivation is not altered by hypoxia, SUMO1 or SENP1.

$-67 \pm 2$ mV, the last parameter a measure of the number of channels available to pass current (**Table 1**). When $O_2$ was lowered from ambient levels to 5% by perfusion of cells with hypoxic solutions (**Plant et al., 2002**), the mean peak $I_{Na}$ increased over 40 s to a new, stable level that was ~70% higher, $-294 \pm 25$ pA/pF (**Figure 1a** and **Table 1**), reminiscent of increases in $I_{Na}$ in response to acute hypoxia reported by others studying rat neurons from the hypothalamus (**Horn and Waldrop, 2000**) and hippocampus (**Raley-Susman et al., 2001**). Augmentation of $I_{Na}$ by hypoxia was associated with a leftward shift of $-11 \pm 2$ mV in both $V_{1/2}$ and SSI, allowing the same amount of depolarization to evoke larger $I_{Na}$ currents (**Figure 1b**). The hypoxia-induced increase in $I_{Na}$ was long-lasting, remaining unchanged 10 min after neurons were restored to ambient $O_2$ (**Supplementary file 1a**). Hypoxia did not alter the kinetics of recovery of $I_{Na}$ from the fast-inactivated state (**Figure 1—figure supplement 1**).

## The SUMO pathway regulates CGN $I_{Na}$

Previously, we demonstrated that SUMO introduced in the recording pipette, or washed past excised inside-out membrane patches, decreased $I_{DR}$ in hippocampal neurons (**Plant et al., 2011**) and suppressed $I_{Kso}$ in CGN (**Plant et al., 2012**) due to SUMOylation of their pore-forming channel α-subunits, $K_V2.1$ and K2P1, respectively. Moreover, we found that the enzymes that mature, activate, and conjugate SUMO to the channels reside on the cytosolic face of the plasma membrane in tissue culture cells and neurons (**Plant et al., 2010**, **2011**, **2012**). Here, seeking evidence for

**Table 1.** Effects of hypoxia, SUMO1 and SENP1 on native $I_{Na}$ and cloned $Na_V1.2$ channels. Neurons (**Figures 1** and **3**) or cloned channels in CHO cells (**Figure 5**) were studied in whole-cell mode. Stimulation protocols are described in the Materials and methods. $V_{1/2}$, the voltage evoking half-maximal conductance; $k$, the slope of the curve were obtained by fitting the normalized current plotted against voltage to a Boltzmann function, $I = I_{max}/(1+\exp[-(V-V_{1/2})/k])$, where $I_{max}$ is maximum current; and SSI is the steady-state inactivation half voltage. For comparison between groups, current densities were measured both at $-20$ mV, to demonstrate the impact of the shifts in $V_{1/2}$, and at 0 mV, a potential where the G-V relationships are saturated under all study conditions. The maximal current in CGN cultured at 21% $O_2$ and studied in ambient $O_2$ was at $-5 \pm 1$ mV (and the current-density was $292 \pm 15$ pA/pF); when these neurons were studied with SUMO1 in the pipette or subjected to 5% $O_2$, the maximal current was observed at $-20 \pm 2$ mV and $-20 \pm 3$ mV, respectively, shifts of ~15 mV analogous to those seen in $V_{1/2}$. When the cells were studied with SENP1 in the pipette, the maximal current was measured at $+5 \pm 2$ mV, a shift of ~10 mV, and the current-density was $-288 \pm 17$ pA/pF. Data are means $\pm$ S.E.M. for 10 to 15 cells per group; * indicates $p<0.05$ compared with cells studied at ambient $O_2$ under control conditions.

| | $I_{Na}$ | | | | | | $Na_V1.2$ | | | | | | $Na_V1.2$-Lys38Gln | | | | |
| | Activation | | SSI | | $I_{-20 mV}$ | $I_{0 mV}$ | Activation | | SSI | | $I_{-20 mV}$ | $I_{0 mV}$ | Activation | | SSI | | $I_{-20 mV}$ |
| | $V_{1/2}$ mV | k | $V_{1/2}$ mV | k | pA/pF | pA/pF | $V_{1/2}$ mV | k | $V_{1/2}$ mV | k | pA/pF | pA/pF | $V_{1/2}$ mV | k | $V_{1/2}$ | k | pA/pF |
|---|---|---|---|---|---|---|---|---|---|---|---|---|---|---|---|---|---|
| Cultured at 21% $O_2$ | $-23 \pm 0.5$ | $4.0 \pm 0.2$ | $-67 \pm 2$ | $6 \pm 1$ | $-172 \pm 20$ | $-293 \pm 12$ | $-18.7 \pm 0.2$ | $3.7 \pm 0.1$ | $-59 \pm 0.4$ | $7 \pm 2$ | $-112 \pm 8$ | $-194 \pm 10$ | $-4.0 \pm 1.5$ | $3.6 \pm 0.3$ | $-50 \pm 1$ | $6.5 \pm 1$ | $-33 \pm 7$ |
| Lowering $O_2$ 21% to 5% | $-34 \pm 1.5^*$ | $4.2 \pm 0.2$ | $-78 \pm 2^*$ | $10 \pm 2$ | $-294 \pm 25^*$ | $-287 \pm 18$ | $-30 \pm 0.5^*$ | $4 \pm 1$ | $-72 \pm 1.5^*$ | $7.5 \pm 1$ | $-192 \pm 11^*$ | $-187 \pm 9$ | $-4.0 \pm 2$ | $3 \pm 1$ | $-49 \pm 1$ | $6 \pm 1$ | $-35 \pm 6^*$ |
| SENP1 | $-7.5 \pm 1^*$ | $4.1 \pm 0.3$ | $-53 \pm 1^*$ | $8 \pm 2$ | $-42 \pm 12^*$ | $-285 \pm 17$ | $-2.5 \pm 0.3^*$ | $3.6 \pm 0.1$ | $-48 \pm 0.5^*$ | $6 \pm 1$ | $-29 \pm 9^*$ | $-191 \pm 11$ | $-3.5 \pm 1$ | $3.2 \pm 0.4$ | $-49 \pm 0.5$ | $6.5 \pm 2$ | $-35 \pm 7^*$ |
| SUMO1 | $-36 \pm 1^*$ | $3.9 \pm 0.2$ | $-77 \pm 3^*$ | $8.5 \pm 1$ | $-303 \pm 17^*$ | $-289 \pm 15$ | $-30 \pm 0.3^*$ | $3.5 \pm 0.2$ | $-69 \pm 0.5^*$ | $7.5 \pm 2$ | $-196 \pm 17^*$ | $-193 \pm 12$ | $-4.0 \pm 2$ | $3.4 \pm 0.5$ | $-50 \pm 1.5$ | $6 \pm 2$ | $-33 \pm 5^*$ |
| Cultured at 7% $O_2$ | $-42.9 \pm 1.5$ | $3.5 \pm 0.5$ | $-57 \pm 3$ | $7 \pm 2$ | $-162 \pm 12$ | | N.D. | | | | | | | | | | |
| Lowering $O_2$ 7% to 1.5% | $-49.5 \pm 1.0$ | $3.2 \pm 0.8$ | $-69 \pm 2$ | $7 \pm 2$ | $-267 \pm 18$ | | N.D. | | | | | | | | | | |

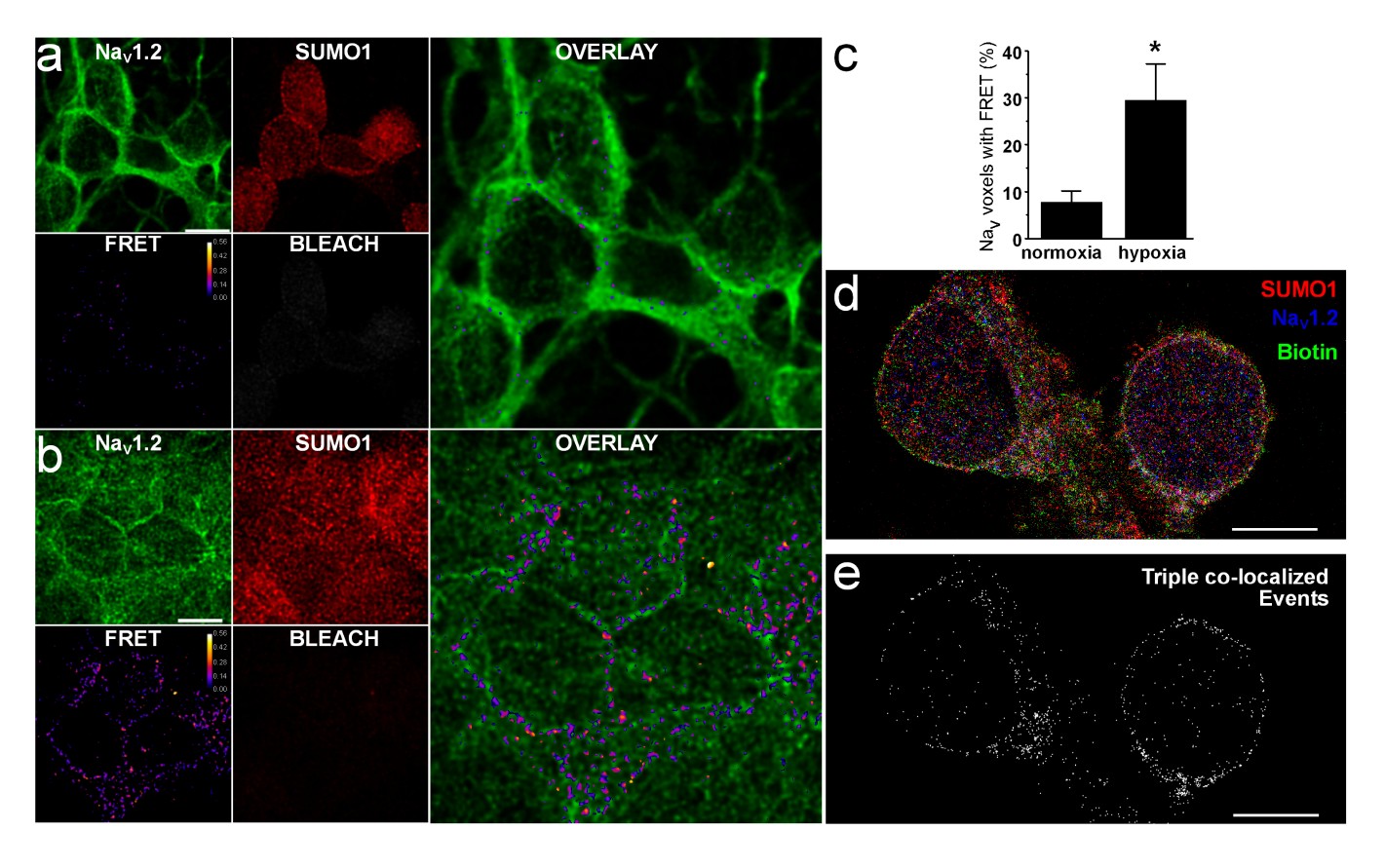

**Figure 2.** $Na_V1.2$ assembles with native SUMO1 at the surface in rat CGN. The association of native $Na_V1.2$ and SUMO1 in CGN was studied by amFRET and STORM per the Materials and methods. Data are mean ± S.E.M. The scale bar represents 5 µm. (**a**) Representative photomontage of amFRET of $Na_V1.2$ and SUMO in cells exposed to normoxia, using Alexa Fluor 488 and Alexa Fluor 594-labeled secondary antibodies to anti-$Na_V1.2$ and anti-SUMO1. The four smaller images show $Na_V1.2$ (donor, top left), SUMO1 (acceptor, top middle), the acceptor after photobleaching (BLEACH, bottom right), and FRET (bottom left, FRET efficiency indicated in pseudo-colored scale); the calculated FRET ratio in voxels after acceptor photobleaching was 0.15 ± 0.01 (n = 10 neurons on six coverslips). The large panel is an overlay of the donor and resultant FRET. (**b**) Representative photomontage of amFRET of $Na_V1.2$ and SUMO in cells exposed to hypoxia (1% $O_2$), using the identical approach and image layout as in panel **a** showing a calculated FRET ratio after acceptor photobleaching of 0.28 ± 0.05 (n = 8 neurons on five coverslips). (**c**) Histogram showing a four-fold increase in the voxels with $Na_V1.2$ demonstrating FRET from ambient conditions (7.8 ± 2.3%) with hypoxia (29.6 ± 7.6%, n = 8 on five coverslips); * indicates p<0.001. (**d**) Composite scatterplot of fluorophore localizations obtained with STORM imaging. Maximum z-projections of scatterplots of fluorophore events from Alexa 568-labeled anti-SUMO1 antibodies (red), Alexa 647-labeled anti-$Na_V1.2$ antibodies, and Alexa 488-tagged streptavidin bound to extracellular membrane proteins are superimposed. Pixels are 20 nm square. (**e**) Mask of 20 nm square voxels in which fluorophore events from anti-SUMO1, anti-$Na_V1.2$ and streptavidin co-localize for exemplar neurons (n = 5).

The following figure supplement is available for figure 2:

**Figure supplement 1.** FRET was not observed between native $Na_V1.2$ and GAD67 in CGN.

regulation of $I_{Na}$ by the SUMO pathway, we delivered 100 pm SUMO1 into CGN via the patch-pipette, a concentration that produces maximal effects on the $K^+$ channels. We observed SUMO-induced changes in $I_{Na}$ like those produced by acute hypoxia: peak $I_{Na}$ increased by ~76% and $V_{1/2}$ and SSI were left-shifted by −13 ± 1 mV and −10 ± 3 mV, respectively (*Figure 1c,d* and *Table 1*). Also like hypoxia, SUMO1 did not change the kinetics of recovery from fast inactivation (*Figure 1— figure supplement 1*).

Because SUMO1 increased $I_{Na}$, we anticipated that conditions favoring lysis of SUMO-target adducts would decrease the current. Indeed, intracellular application of 250 pm SENP1 deSUMOy-lase via the pipette suppressed peak $I_{Na}$ by ~75% to −42 ± 12 pA/pF and right-shifted $V_{1/2}$ and SSI by

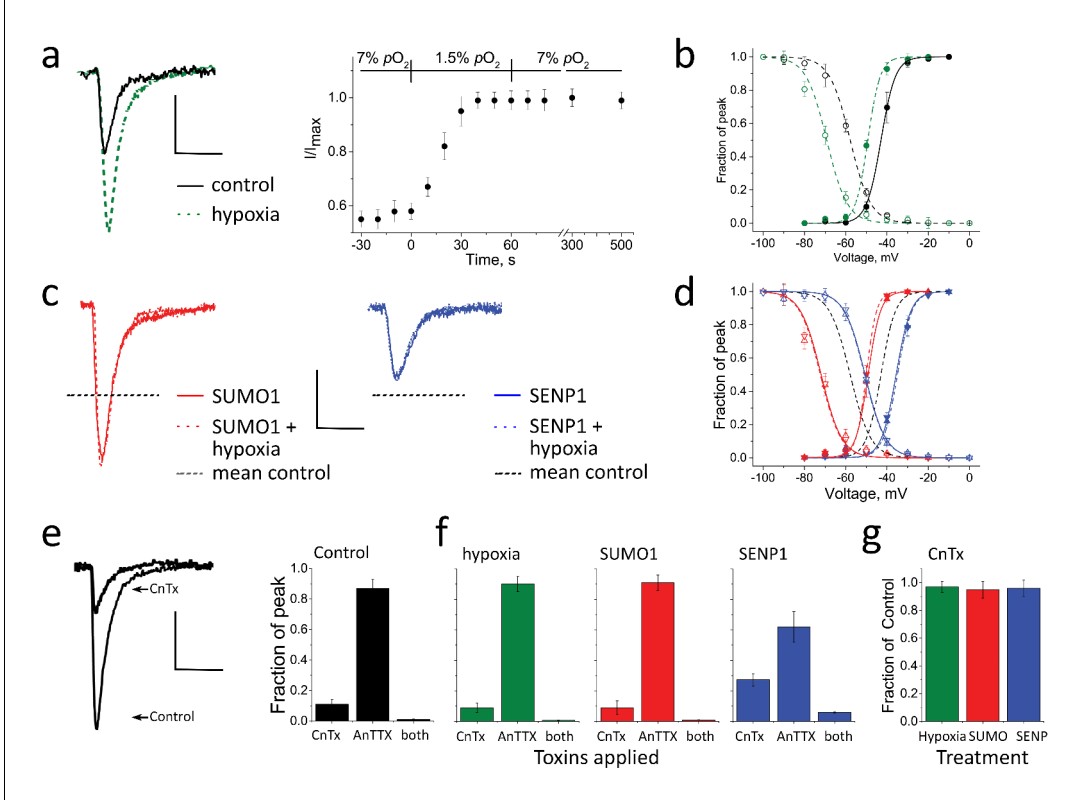

**Figure 3.** Acute hypoxia at 1.5% $O_2$ after culturing at 7% augments $I_{Na}$. $I_{Na}$ in rat CGN was studied following 5–7 days in culture at 7% $O_2$ by whole-cell patch clamp, as per *Figure 1*. Normalized activation (Act) and steady-state inactivation (SSI) relationships were obtained and fit as described in the Materials and methods. Measured values are noted in the text and listed in *Supplementary file 1b*. Cells were studied with a control solution (black), 100 pm SUMO1 (red), or 250 pm SENP1 (blue) in the recording pipette. Sensitivity to CnTx and AnTTX was assessed as per *Figure 1*. Data are mean ± S.E.M. for six cells per group. Scale bars are 150 pA/pF and 5 ms. (a) Left, example traces showing $I_{Na}$ at −20 mV increased when perfusate at 7% $O_2$ (black) was exchanged with a hypoxic solution at 1.5% $O_2$ (green dashes). Right, the time-course for changes in peak current in response to decreased $O_2$ from 7 to 1.5%, normalized to the maximal current for each cell studied. (b) Hypoxic solution (green dashes) left shifted the $V_{½}$ of $I_{Na}$ from control (black) for both Act (solid) and SSI (open). (c) Left, SUMO1 in the pipette (red) increased $I_{Na}$ and no further augmentation was observed by subsequent hypoxia (red dashes). Right, SENP1 in the pipette (blue) decreased $I_{Na}$ and suppressed the response to hypoxia (blue dashes). Dotted black lines indicate the mean peak values obtained at 7% $O_2$ in a. (d) SUMO1 left shifted the $V_{½}$ of Act (red triangle) and SSI (open red triangle). The relationships were then insensitive to hypoxia (Act, red down triangle; SSI, open red down triangle). SENP1 right shifted the $V_{½}$ of Act (blue triangle) and SSI (open blue triangle) and currents were then insensitive to hypoxia. Dashed black lines indicate mean peak values with control solutions from b. (e) Left; $I_{Na}$ studied at 7% $O_2$ then with toxins in the bath. Right; mean normalized peak current histograms showing 89 ± 1.2% inhibition by CnTx, 12 ± 2% inhibition by AnTTX and 99 ± 0.8% inhibition by both toxins (black bars). These ratios indicate that the relative contribution of $Na_V1.2$ and $Na_V1.6$ to $I_{Na}$ is not altered when cells are cultured at 7% $O_2$. (f) Mean normalized peak $I_{Na}$ histograms. Left; 91 ± 1% inhibition by CnTx and 10 ± 1% inhibition by AnTTX with a drop to 1.5% $O_2$ for 60 s (green). Middle; 90 ± 2% inhibition by CnTx and 9 ± 2% inhibition with 100 pm SUMO1 in the pipette (red). When SENP1 was included in the recording pipette (blue), $I_{Na}$ was inhibited by 73 ± 3% by CnTx, 38 ± 8% by AnTTX and 95 ± 2% by both toxins consistent with the passage of much of the remaining current by $Na_V1.6$. (g) Mean normalized peak current histograms show that the $I_{Na}$ remaining after inhibition of $Na_V1.2$ by CnTx did not respond to acute hypoxia at 1.5% $O_2$ (green), SUMO1 (red), or SENP1 (blue).

15 ± 1 mV and 14 ± 2 mV, respectively (*Figure 1c,d* and *Table 1*). Providing additional evidence that hypoxia acted on $I_{Na}$ via the SUMO pathway, currents increased by SUMO1 were not further augmented by hypoxia, nor did hypoxia overcome suppression of $I_{Na}$ by SENP1 (*Figure 1c,d* and *Supplementary file 1a*).

The full excursion in $V_{½}$ between SUMOylating and deSUMOylating conditions was 26.5 ± 3.0 mV (*Figure 1d* and *Table 1*). Supporting the notion that rapid changes in $I_{Na}$ were due to shifts in the conductance-voltage (G-V) relationship, rather than changes in the number of channels on the surface passing current or unitary channel conductance, maximal currents at 0 mV, a voltage where G-V relationships were saturated under all test conditions (*Figure 1b and d*), were not significantly

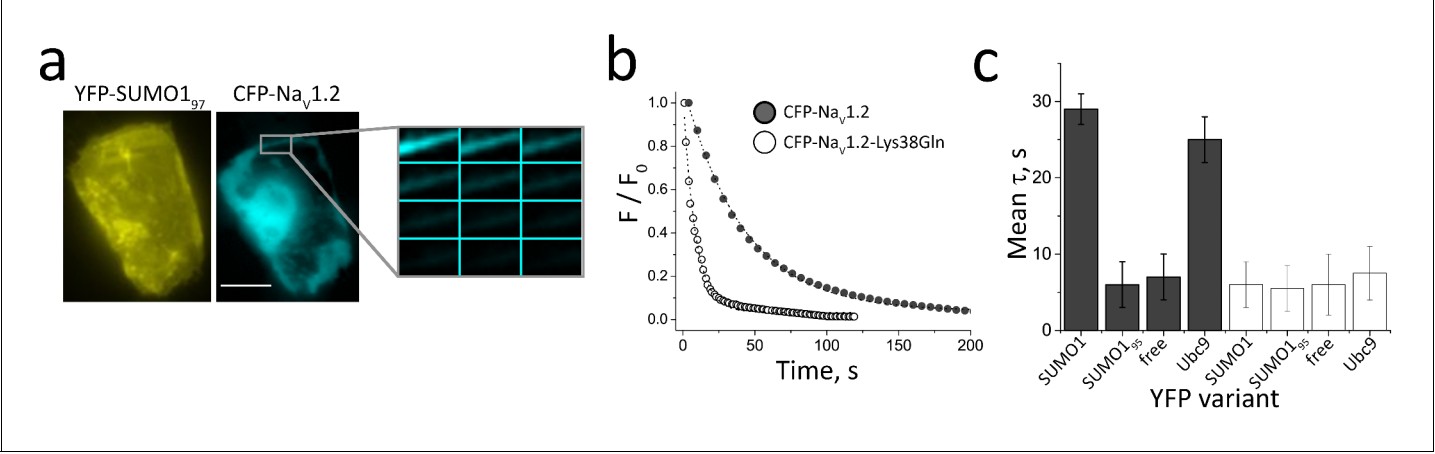

**Figure 4.** FRET between $Na_V1.2$ and SUMO1 at the cell surface requires Lys38. Rat $Na_V1.2$ was expressed in CHO cells with the $\beta1$ subunit and studied in live cells per Materials and methods. FRET was assessed by measuring the time constant ($\tau$) for CFP-photobleaching (donor) in the presence of YFP (acceptor) from 3 regions of 5–7 cells per group. Data are mean $\tau$ ± S.E.M. (a) CFP-$Na_V1.2$ subunits (blue) and YFP-SUMO1 (yellow) reach the cell surface. Scale bar is 10 μm. (b) Exemplar photobleaching studies show the decay of fluorescence intensity for single cells expressing CFP-$Na_V1.2$ (open) or CFP-$Na_V1.2$-Lys38Gln (solid) with YFP-SUMO1 fit by an exponential to give $\tau$. (c) FRET shows the assembly of CFP-$Na_V1.2$ (grey bars) with YFP-SUMO1 and YFP-Ubc9 ($\tau$ = 29 ± 2* and 25 ± 3*, respectively) but not with linkage-incompetent YFP-SUMO1_{95} or free YFP ($\tau$ = 6 ± 3 and 7 ± 3, respectively). In contrast, CFP-$Na_V1.2$-Lys38Gln (white bars) did not show FRET with YFP-SUMO1, YFP-SUMO1_{95}, YFP-Ubc9 or free YFP ($\tau$ = 6 ± 3; 5.5 ± 3.5; 7.5 ± 3.5; and 6 ± 4, respectively). Significant changes in $\tau$ compared to free YFP are indicated (*, p<0.001).

different for CGN at 21% $O_2$ with the three pipette solutions (i.e. control, SUMO1, or SENP1) or on acute exposure to 5% $O_2$ (*Table 1*). Indicating that the SUMO pathway exerted tonic control over $I_{Na}$ under normoxic conditions, CGN studied with control pipette solution rested midway between the $V_{1/2}$ extremes. Shifts in $V_{1/2}$ were not associated with changes in the slope of the G-V relationship (k, *Table 1*).

## $Na_V1.2$ channels conduct the portion of CGN $I_{Na}$ responsive to hypoxia and the SUMO pathway

$I_{Na}$ is passed mainly by $Na_V1.2$ in rat CGN with smaller contributions by other channel isoforms, $Na_V1.2 >> Na_V1.6 >> Na_V1.3$ and $Na_V1.1$ (*Schaller and Caldwell, 2003*; *Shah et al., 2001*). In keeping with this expression pattern, 250 n**m** CnTX, a toxin that blocks $Na_V1.2$ ~100-fold more effectively than $Na_V1.6$ (*Wilson et al., 2011*), inhibited ~90% of $I_{Na}$ at −20 mV in CGN (*Figure 1e*). When CGN were subjected to acute hypoxia, or SUMO1 was applied in the pipette, CnTX still blocked over 90% of $I_{Na}$, indicating that $Na_V1.2$ remained the primary contributor to $Na^+$ current under these conditions (*Figure 1f*). In contrast, the residual current after CnTX blockade was insensitive to acute hypoxia, SUMO1 and SENP1 (*Figure 1g*), suggesting that other $Na_V$ channels did not contribute significantly to the hypoxic response.

Supporting the notion that most of the residual current is passed by $Na_V1.6$, 50 nm 4,9 anhydro-tetrodotoxin (AnTTX), a toxin that blocks $Na_V1.6$ ~150-fold more effectively than $Na_V1.2$ (*Rosker et al., 2007*), inhibited ~10% of $I_{Na}$ under control conditions and >90% of the current remaining after CnTX application (*Figure 1e,f*). Moreover, with SENP1 in the pipette, the fraction of $I_{Na}$ blocked by CnTX was decreased and suppression by AnTTX was increased (*Figure 1h*), the expected response if deSUMOylation decreased the activity of $Na_V1.2$ channels so that $Na_V1.6$ channels passed a larger fraction of the current. When combined, CnTX and AnTTX blocked >95% of $I_{Na}$ in all CGN studied. Taken together, the toxin data are consistent with the conclusion that hypoxia, SUMO1 and SENP1 act to alter the activity of native $Na_V1.2$ channels in CGN.

## Interaction of native Na$_V$1.2 and SUMO1 at the CGN surface increases with hypoxia

To confirm the association of native Na$_V$1.2 and SUMO1 at the rat CGN surface we employed amFRET (*Figure 2a–c*), a method we used previously to demonstrate SUMOylation of native K$^+$ channel α-subunits at the plasma membrane in rat hippocampal neurons and CGN (*Plant et al., 2011*, *2012*). Whereas conventional confocal microscopy identifies two proteins as co-localized in the X-Y plane when they are separated by up to 200 nm, and is more accurately an indication of proximity, FRET reports on short distances between fluorophores (1–10 nm) that are indicative of molecular interaction. Here, Na$_V$1.2 and SUMO1 were labeled with specific primary antibodies and visualized with secondary antibodies carrying donor and acceptor probes, respectively, after neurons were fixed under normoxic or hypoxic conditions.

Many voxels exhibiting Na$_V$1.2 immunoreactivity (donor) showed an increase in fluorescence following bleaching of the acceptor (SUMO), demonstrating FRET between the fluorophores (*Figure 2a–c*). Thus, in ambient O$_2$, 7.8 ± 2.3% of Na$_V$1.2 voxels showed FRET (*Figure 2a*, n = 10 neurons), whereas in neurons exposed to 1% O$_2$ for 5 min, the percentage of voxels showing FRET increased ~4-fold to 29.6 ± 7.6% (*Figure 2b*, n = 8 neurons). Supporting specificity of signals generated between Na$_V$1.2 and SUMO1, no FRET was observed between Na$_V$1.2 and GAD67, an abundant cytosolic protein (*Figure 2—figure supplement 1*).

Because the CGN nucleus occupies much of the intracellular volume of the soma, its separation from the plasma membrane is often uncertain. To demonstrate directly that Na$_V$1.2 was interacting with SUMO1 at the neuronal surface, we tagged the extracellular portions of plasma membrane proteins with biotin to allow visualization with Alexa488-streptavidin and imaged Na$_V$1.2 and SUMO1 with Alexa Fluor 647 and Alexa Fluor 568 labeled secondary antibodies, respectively. Using STORM, a technique that offers a resolution in the X-Y plane of ≤20 nm, we localized single molecules of the three fluorophores (*Figure 2c*) and then identified voxels in which single fluorescent events from all three fluorophores were observed (*Figure 2d*). Triple co-localization delineated the circumference of the neuronal soma (n = 5), consistent with plasma membrane localization of SUMOylated Na$_V$1.2, as predicted by the electrophysiological studies.

## CGN cultured at 7% O$_2$

The neuronal response to acute hypoxia is often studied in cells cultured at ambient levels of oxygen. However, the baseline O$_2$ level in mammalian brain tissue has been estimated to be as low as ~7% under normal, healthy conditions and to drop to under 2% with ischemia (*Ponce et al., 2012*; *Lyons et al., 2016*). Therefore, we sought to reevaluate $I_{Na}$ responses at O$_2$ concentrations more reflective of in vivo conditions. The responses of $I_{Na}$ to acute hypoxia, SUMO and SENP were studied at lower O$_2$ concentrations as follows (*Figure 3*, *Table 1*, and *Supplementary file 1b*). First, rat CGN were confirmed to tolerate long-term culture in 7% O$_2$, as we previously observed for rat hippocampal neurons (*Shelat et al., 2013*). Next, CGN cultured for 7–10 days at 7% O$_2$ were evaluated at 7% O$_2$ and then on exposure to 1.5% O$_2$; the decrease in O$_2$ led to an acute increase in mean peak $I_{Na}$ of ~65% and, starting from a baseline for $V_{1/2}$ of −42 ± 1.5 mV and −57 ± 3 mV, $V_{1/2}$ and SSI were left-shifted by −8 ± 2 mV and −11 ± 3 mV, respectively. Similarly, SUMO1 increased $I_{Na}$ in CGN cultured at 7% O$_2$ by ~68%, in association with left-shifts in $V_{1/2}$ and SSI of − 8 ± 1.5 mV and −12 ± 3 mV, respectively, while SENP1 suppressed $I_{Na}$ by ~70% and right-shifted $V_{1/2}$ and SSI by 9 ± 1 mV and 8 ± 2 mV, respectively. As observed when CGN were cultured in ambient O$_2$, the increase in $I_{Na}$ induced by SUMO1 when cells were incubated at 7% O$_2$ was not further augmented when O$_2$ was decreased to 1.5% nor could subsequent exposure to 1.5% O$_2$ overcome suppression of $I_{Na}$ by SENP1. Finally, like neurons cultured in ambient O$_2$, blockade studies with CnTX and AnTTX supported the identification of Na$_V$1.2 as the channel passing hypoxia-responsive $I_{Na}$ in CGN cultured at 7% O$_2$, and the proposal that Na$_V$1.6 is responsible for the residual current that was insensitive to hypoxia, SUMO1 or SENP1 (*Figure 3e–g*).

## SUMOylation of Na$_V$1.2 on Lys38 is necessary and sufficient for the hypoxic response

One Lys in the 2005 amino acid, pore-forming Na$_V$1.2 channel α-subunit was shown to be subject to SUMOylation, and to mediate the response to acute hypoxic challenge, using heterologous

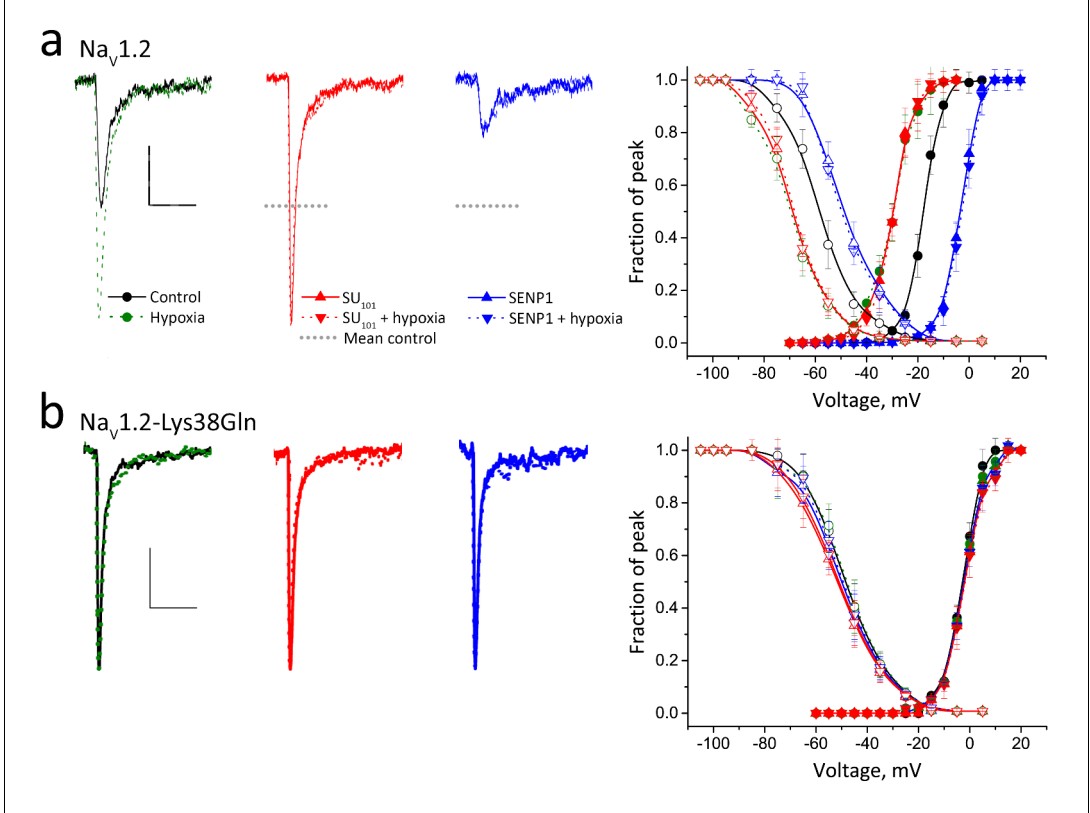

**Figure 5.** Acute hypoxia induces SUMOylation of $Na_V1.2$ on Lys38. Rat $Na_V1.2$ was expressed in CHO cells with the $\beta1$ subunit and studied under normoxic and hypoxic conditions with control solution (black), 100 pm SUMO1 (red) or 250 pm SENP (blue) in the recording pipette, as indicated. Data are mean ± S.E.M. for 10 to 15 cells per group. Measured values are noted in the text and listed in *Table 1*. Scale bars are 5 ms and 50 pA/pF in **a** and 10 pA/pF in **b**. (**a**) Left, example traces show hypoxia (green dashes) increased $Na_V1.2$ channel current from control conditions (black). SUMO1 in the pipette (red) increased the current to the same level and precluded further augmentation by hypoxia (red dashes). In contrast, SENP1 (blue) decreased the current by ~75% and suppressed sensitivity to hypoxia (blue dashes). Dotted black lines represent mean peak current under control conditions. Right, both Act (solid) and SSI (open) for $Na_V1.2$ were left shifted when cells with hypoxia (green) or SUMO1 in the pipette (red) and were right-shifted with SENP in the pipette (blue). Hypoxia caused no further change to Act or SSI with SUMO1 or SENP1 in the pipette (dashed lines). (**b**) Left, $Na_V1.2$-Lys38Gln channels passed smaller currents (black) that were not increased by hypoxia (green) or by SUMO1 in the pipette (red) or decreased by SENP1 (blue) in the pipette. Right, the normalized Act and SSI relationships for $Na_V1.2$-Lys38Gln channels (black) are right-shifted compared to wild type $Na_V1.2$ channels and do not change when cells are exposed to hypoxia (green) or are studied with SUMO1 (red) or SENP (blue) in the pipette.

expression of wild type and mutant proteins in CHO cells. This reconstitution strategy, previously exploited to study SUMOylation of K[+] channels, included here co-expression of the wild type human $Na_V$ $\beta1$ subunit in all cells unless otherwise noted.

First, we sought to confirm the association of $Na_V1.2$ and SUMO1 at the CHO cell surface by measuring FRET between subunits tagged with Yellow and Cyan FPs (YFP and CFP), a method we used before with K[+] channels (*Rajan et al., 2005*; *Plant et al., 2010*, *2011*, *2012*). Tagging $Na_V1.2$ with CFP (CFP-$Na_V1.2$) did not alter the biophysical properties of the channel nor its responsiveness to hypoxia, SUMO1, or SENP1 (*Supplementary file 1a*). As we previously reported, tagging the N-terminus of SUMO1 with FPs does not alter their operation, and this was observed here with $Na_V1.2$ (*Rajan et al., 2005*; *Plant et al., 2010*, *2011*, *2012*). Both YFP-SUMO1 and CFP-$Na_V1.2$ were visualized at the plasma membrane when expressed together in CHO cells (*Figure 4a*); of note, unlike CGN, the volume of cytosol in CHO cells allows ready visualization of the cell surface in regions distinct from the nucleus. Consistent with SUMOylation of CFP-$Na_V1.2$, the time course of CFP decay with continuous illumination, as assessed by the time constant ($\tau$) for donor photobleaching, was prolonged from under 10 s to ~30 s when the channel was expressed with YFP-SUMO1 (*Figure 4b,c*). CFP-$Na_V1.2$ also showed FRET with YFP-Ubc9, the E2 SUMO-conjugating enzyme

that binds to its targets only when the Lys residues subject to modification are present (*Plant et al., 2011*). In contrast, FRET was not observed on expression of CFP-Na$_V$1.2 with linkage-incompetent YFP-SUMO1$_{95}$ (a variant that lacks the terminal Gly-Gly motif present in mature SUMO1$_{97}$) or soluble YFP; in both cases, $\tau$ was less than 10 s.

FRET was used to demonstrate that Lys38 in the cytoplasmic N-terminus of the Na$_V$1.2 $\alpha$-subunit was necessary and sufficient to mediate SUMOylation of the channel complex. Among 133 Lys residues in the Na$_V$1.2 $\alpha$-subunit, a sensitive predictive tool, GPS-SUMO (*Zhao et al., 2014*), identified Lys38 as one of just five potential SUMOylation sites. Mutating Lys38 to Gln, removing the side chain subject to SUMO modification, produced CFP-Na$_V$1.2-Lys38Gln subunits that reached the CHO cell surface like wild type CFP-Na$_V$1.2. However, in contrast with CFP-Na$_V$1.2, the mutant failed to exhibit FRET with either YFP-SUMO1 or YFP-Ubc9. The absence of FRET supported the conclusion that Lys38 was required for SUMOylation of the Na$_V$1.2 channel complex and suggested that no other Lys in the $\alpha$ subunit or the $\beta$1 subunit was subject to SUMOylation.

Next, we observed that acute hypoxia altered the function of wild type Na$_V$1.2 channels expressed in CHO cells in the same manner as seen for native $I_{Na}$ in CGN (*Figure 5a* and *Table 1*). Thus, in ambient O$_2$ the biophysical attributes of Na$_V$1.2 channels in CHO cells were like those of CGN $I_{Na}$, showing similar values for $V_{\frac{1}{2}}$ and SSI. Moreover, Na$_V$1.2 channels responded to acute hypoxia with an increase in mean peak current-density of ~70% in less than 40 s in association with leftward shifts in $V_{\frac{1}{2}}$ and SSI of $-11 \pm 1$ mV and $-13 \pm 2$ mV, respectively. Na$_V$1.2 channels in CHO cells also responded to SUMO1 and SENP1 like native $I_{Na}$. Thus, SUMO1 in the pipette increased current-density and left-shifted $V_{\frac{1}{2}}$ and SSI to the same extent as acute hypoxia and SENP1 decreased current-density and right-shifted the $V_{\frac{1}{2}}$ and SSI from baseline by ~75%. The total excursion in $V_{\frac{1}{2}}$ for Na$_V$1.2 channels in CHO cells between these treatments favoring SUMOylation and deSUMOylation was $27.5 \pm 1.0$ mV, as it was for CGN $I_{Na}$. Moreover, the conclusion drawn from native $I_{Na}$ that shifts in the G-V relationships were the primary determinants of acute changes in current density were supported as well for Na$_V$1.2 channels in CHO cells by measurement of similar maximal currents at 0 mV for the various test conditions. Also as observed for CGN $I_{Na}$, acute hypoxia did not further augment Na$_V$1.2 channel currents first increased by SUMO1 nor did it abrogate prior SENP1-mediated current suppression.

Supporting the essential role of SUMOylation of Lys38 in the hypoxic response, expression of Na$_V$1.2-Lys38Gln channels produced currents that were insensitive to acute hypoxia, SUMO1, and SENP1 (*Figure 5b* and *Table 1*). The current-density, $V_{\frac{1}{2}}$ and SSI values for Na$_V$1.2-Lys38Gln channels under all three conditions were like those measured for wild-type Na$_V$1.2 channels subjected to deSUMOylation by SENP1. Like CGN $I_{Na}$ and wild-type Na$_V$1.2 in CHO cells, Na$_V$1.2-Lys38Gln channels recovered from fast inactivation in a fashion that was insensitive to acute hypoxia, SUMO1, or SENP1 (*Figure 1—figure supplement 1*).

Of note, the $\beta$1 subunit did not appear to play a role in the hypoxic response of Na$_V$1.2 via Lys38 or to the regulation of channel activity by the SUMO pathway. Expression of the $\alpha$ subunits without $\beta$1 did not modify the responses of Na$_V$1.2 channels to acute hypoxia, SUMO1, and SENP1 nor did it alter insensitivity of Na$_V$1.2-Lys38Gln channels to the same three manipulations (*Supplementary file 1c*).

Lys38 is in a sequence motif, Pro-Lys38-Gln-Glu (*Figure 6a*), which is conserved near the N-terminus of Na$_V$1.2 from zebrafish to humans. To verify SUMOylation of Lys38 in the motif, we used a mass spectrometry (MS) strategy that we previously developed for K$^+$ channels (*Plant et al., 2010, 2011*). SUMO1$_{97}$T95K, a mutant that leaves the Gly-Gly remnant of SUMO1 on the $\varepsilon$-amino group of target Lys after trypsin treatment was expressed in bacteria with rat Na$_V$1.2 residues 1–125 bearing a His$_6$ affinity tag and the mammalian E1 and E2 SUMOylation enzymes. Affinity purification followed by SDS-PAGE revealed a band of the expected apparent mass for the Na$_V$1.2-SUMO adduct (~40 kDa) that was excised, treated with trypsin and subjected to MS yielding sequence coverage for Na$_V$1.2$_{1-125}$ of 85% (*Figure 6b*). The spectrum included a peak at the predicted size for a fragment carrying Lys38 linked via its side-chain to the Gly-Gly SUMO1 remnant, and MS-MS sequencing identified it uniquely to be the three-ended product (*Figure 6c,d*).

## Hypoxia induces rapid monoSUMOylation of Na$_V$1.2 channels

To determine the stoichiometry of SUMOylation, we counted the number of SUMO1 monomers on individual Na$_V$1.2 channels at the surface of live CHO cells using simultaneous, two-color TIRF microscopy and photobleaching, as before with K$^+$ channels (*Plant et al., 2010, 2011, 2012*). When

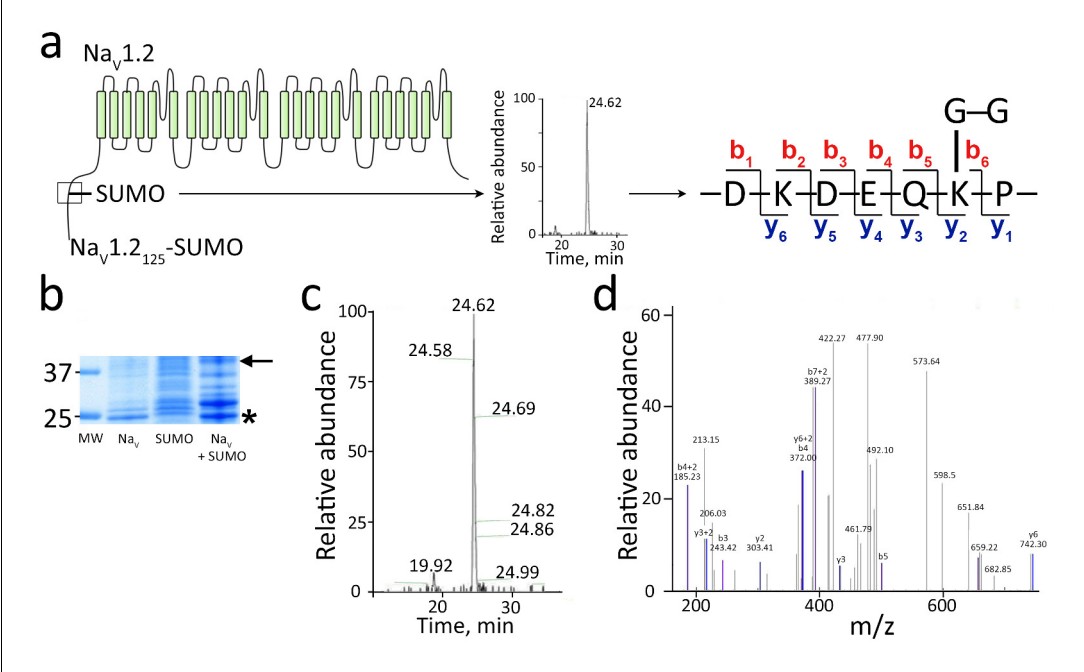

**Figure 6.** Mass spectroscopy shows SUMO1 conjugated to $Na_V1.2$-Lys38. For MS analysis, rat $Na_V1.2_{1-125}$ and $SUMO1_{97}T95K$ were expressed with mouse SUMOylation enzymes in *E. coli*, purified, subjected to trypsin cleavage, and analyzed by MS as described in the Materials and methods. (**a**) Left, schematic showing the product $Na_V1.2_{1-125}$-SUMO (box) and the trypsin fragment of $Na_V1.2$ carrying the Gly-Gly remnant of $SUMO1_{97}T95K$. Right, The sequence of the three-ended fragment with Lys38 and the Gly-Gly remnant. (**b**) A Coomassie blue-stained SDS–PAGE gel of the purified products. Unmodified $Na_V1.2_{1-125}$ ($Na_V$) migrates at ~25 kDa (star). Expression of $SUMO1_{97}T95K$ (SUMO) and the SUMO enzymes yields SUMO on the overexpressed target as well as native proteins (**Plant et al., 2011**; **Uchimura et al., 2004**). The $Na_V1.2_{1-125}$-SUMO conjugate migrates at ~40 kDa (arrow). Molecular weight markers are shown (MW). (**c**) Fourier transform mass spectrum after trypsin digestion of $Na_V1.2_{1-125}$-SUMO expressed as relative abundance against time of capture. The predicted fragment of 422.27 Da was captured as a single peak at 24.62 min. (**d**) Tandem MS sequence analysis of the fragment with Lys38 and the Gly-Gly remnant indicating b and y ion species as annotated in **a**.

CFP-$Na_V1.2$ and SUMO1 tagged with mCherry (mCherry-SUMO1) were co-expressed, particles containing both CFP-$Na_V1.2$ and mCherry-SUMO1 were observed at the plasma membrane. In response to photobleaching, these particles showed only a single bleaching step for each fluorophore under both normoxic and hypoxic conditions, indicating the presence of one fluorescent subunit of each type (**Figure 7a,b** and **Table 2**). Acute hypoxia increased the fraction of CFP-$Na_V1.2$ particles with mCherry-SUMO1 ~4-fold (from 20% to 80%), as expected due to hypoxia-induced SUMOylation (**Figure 7b**). In contrast, SENP1 in the recording pipette suppressed both baseline and hypoxia-induced SUMOylation of CFP-$Na_V1.2$ by mCherry-SUMO1 so that colocalized particles were decreased (**Figure 7c**). Similarly, expression of CFP-$Na_V1.2$-Lys38Gln and mCherry-SUMO1 rarely yielded colocalization of the fluorophores in both ambient $O_2$ and hypoxic conditions (**Figure 7d**). Because channels are formed with one $Na_V1.2$ subunit, observing only a single mCherry-SUMO1 in particles as well confirmed three conclusions: only Lys38 on the α-subunit was SUMOylated; poly-SUMO1 chains were not formed; and, the β1 subunit was not SUMOylated.

Supporting the conclusion that changes in $O_2$ lead to rapid SUMOylation of $Na_V1.2$, acute hypoxia recruited mCherry-SUMO1 to the cell surface only at sites where CFP-$Na_V1.2$ channels were already present and without altering the number of channels at the surface (**Figure 8a,b** and **Table 2**). Here, surface density and localization of CFP-$Na_V1.2$ and mCherry-SUMO1 subunits were assessed using TIRF and pixel-by-pixel analysis at room temperature (**Manders et al., 1993**), a method we have applied to quantify the surface density and co-assembly of cardiac $K^+$ channel α and β subunits (**Plant et al., 2014**). In ambient $O_2$, just 16% of channel pixels were localized with mCherry-SUMO1 ($67 \pm 6/\mu m^2$) and 84% were observed to be unmodified ($340 \pm 16/\mu m^2$). After 2 min of hypoxia, an additional 50% of channel pixels were localized with mCherry-SUMO1, increasing

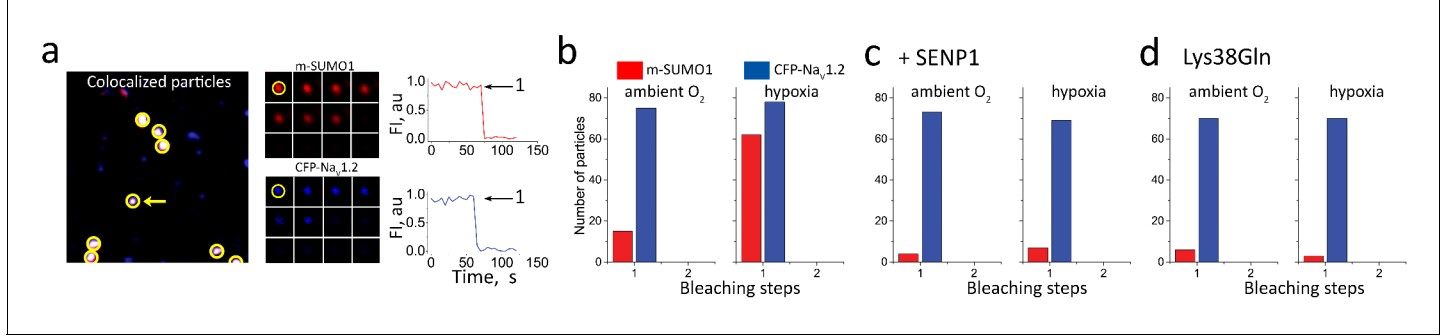

**Figure 7.** Hypoxia recruits one SUMO1 monomer to each cell surface NaV1.2 channel. Single CFP-NaV1.2 or CFP-NaV1.2-Lys38Gln (K38Q, blue) channels and SUMO1 tagged with mCherry (m-SUMO1, red) were studied in CHO cells by TIRFM as described in the Materials and methods. Data represent 5–8 cells in each case and biophysical parameters and single particles statistical analyses are summarized in *Supplementary file 1a* and *Table 2*, respectively. (a) Left, single co-localized particles with both mCherry and CFP fluorescence were observed at the surface of cells expressing NaV1.2 and SUMO1. Simultaneous continuous photobleaching time courses revealed complexes to have one subunit of each type. (b) Histogram of photobleaching steps showing that hypoxia increased single mCherry-SUMO1 (red) subunits at the cell surface co-localized with NaV1.2 channels (blue), without a change in subunit stoichiometry (*Table 2*). (c) Histogram of photobleaching steps showing that SENP suppresses hypoxia-induced increase in single mCherry-SUMO1 (red) subunits at the cell surface co-localized with NaV1.2 (blue). (d) Histogram of photobleaching steps showing that hypoxia does not increase in single mCherry-SUMO1 (red) subunits at the cell surface co-localized with NaV1.2-Lys38Gln channels (blue).

**Table 2.** Co-localization of SUMO1 with NaV1.2 in response to hypoxia. CFP-tagged NaV1.2 or NaV1.2-Lys38Gln subunits were expressed in CHO cells with mCherry-SUMO1 (m-SUMO1) and studied by TIRFM and whole-cell patch-clamp (*Figures 7* and *8*). The number of photobleaching steps observed for each fluorophore in each single fluorescent spot reports on the stoichiometry of the channel complex. NaV1.2 channels are monomers and show no more than one bleaching step when tagged with CFP (*Figure 7*). No more than one bleaching steps was observed for mCherry-tagged SUMO1 subunits (free or co-localized with the channel). A 1:1 stoichiometry is maintained when cells are exposed to hypoxia. SUMO1 was not observed to co-localize with NaV1.2-Lys38Gln channels. The surface density of subunits was quantified as the mean of four 100 by 100 pixel regions for 6–10 cells per group. Exposure to hypoxia increased the number of SUMO1 monomers observed at the cell surface within 40 s and almost all were co-localized with NaV1.2. Whole-cell, peak current-density, measured at −20 mV, increased by ~70% within 40 s of hypoxia and remained stable during 2 min of hypoxia and 20 min of recovery at ambient levels of $O_2$. Pulse protocols to determine the activation (Act) and steady-state inactivation (SSI) $V_{1/2}$ values (the voltage evoking half-maximal conductance) were obtained as described in the Materials and methods and the manuscript Table. Data are means ± S.E.M. for 5 to 8 cells per group; * indicates p<0.05 compared with cells studied in ambient $O_2$ for each channel type studied.

| Subunits expressed | CFP-NaV1.2 + m-SUMO1 | | | | | | | CFP-NaV1.2-Lys38Gln + m-SUMO1 | | |
|---|---|---|---|---|---|---|---|---|---|---|
| Condition | Ambient $O_2$ | Hypoxia 40 s | Hypoxia 2 min | Recovery 5 min | Recovery 10 min | Recovery 20 min | SENP1 | Ambient $O_2$ | Hypoxia 40 s | Recovery 5 min |
| Single particle stoichiometry SUMO1: NaV1.2 | 1: 1 | ND | 1: 1 | 1: 1 | ND | ND | 0: 1 | 0: 1 | 0: 1 | ND |
| Free CFP-NaV1.2 pixels / μm² | 340 ± 16 | 139 ± 8 | 137 ± 11 | 134 ± 7 | 131 ± 6 | 129 ± 8 | 290 ± 10 | 309 + 12 | 303 + 15 | 305 + 12 |
| Free mSUMO1 pixels / μm² | 4 ± 5 | 12 ± 2 | 11 ± 3 | 10 ± 7 | 8 ± 6 | 8 ± 5 | 3 + 2 | 5 + 2 | 4 + 2 | 4 + 1 |
| Co-localized pixels / μm² | 67 ± 6 | 268 ± 12 | 265 ± 12 | 260 ± 11 | 245 ± 14 | 239 ± 8 | 3 + 2 | 2 + 1 | 2 + 2 | 1 + 2 |
| Total CFP-NaV1.2 pixels / μm² | 407 ± 10 | 407 ± 12 | 402 ± 10 | 394 ± 15 | 376 ± 14 | 362 ± 10 | 293 ± 8 | 311 ± 12 | 305 ± 16 | 306 ± 14 |
| Act $V_{1/2}$ (mV) | −22 ± 1.2 | −31 ± 1.7 | −32 ± 2 | −29 + 2 | −35 ± 1.4 | −33 + 2 | −3.5 ± 1.8 | −4 + 2 | −3.7 + 2.5 | −3.1 ± 1.2 |
| SSI $V_{1/2}$ (mV) | −61 ± 2 | −70 ± 3 | −69 ± 4 | −71 + 3 | −70 ± 2 | −68 + 3 | −44 ± 1.5 | −51 + 1.5 | −53 + 3 | −48 ± 2 |
| $I_{Peak}$ (pA/pF) | −120 ± 8 | −198 ± 10 | −199 ± 13 | −200 ± 14 | −201 ± 13 | −189 ± 17 | −33 ± 12 | −39 ± 13 | −37 ± 9 | −39 ± 14 |

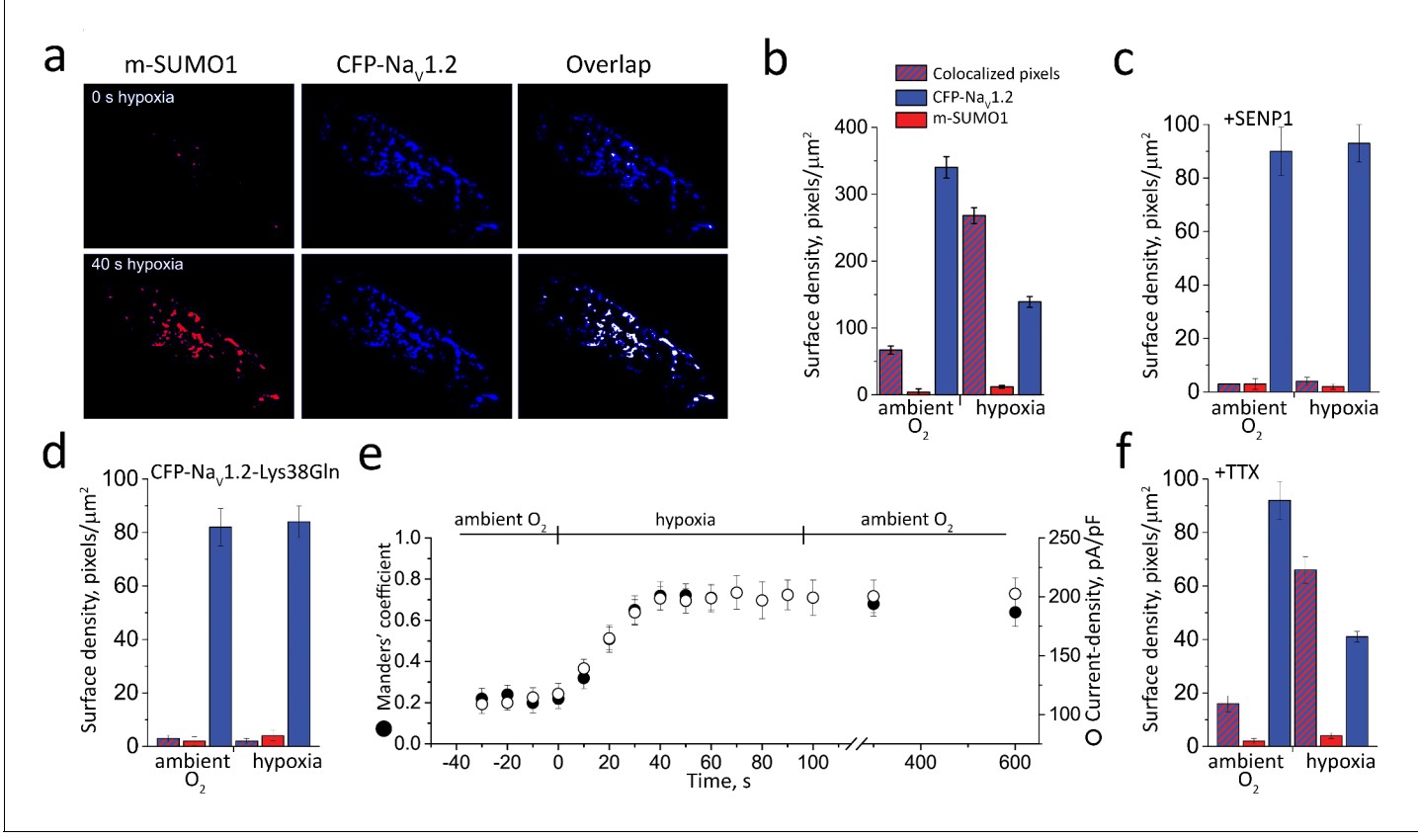

**Figure 8.** Hypoxic SUMOylation of $Na_V1.2$ and current density proceed concurrently. $CFP-Na_V1.2$ or $CFP-Na_V1.2-Lys38Gln$ (K38Q, blue) channels and SUMO1 tagged with mCherry (m-SUMO1, red) were studied in CHO cells by TIRFM and pixel-by-pixel analysis performed as described in the Materials and methods. Briefly, images were captured at 5 s intervals and data for each fluorophore saved as separate stacks; the background was subtracted and processed for misalignment in an identical manner. Manders' coefficients were assessed post-hoc for 3–5 regions per cell. Co-localization was defined as the presence of both fluorophores at more than 30% of the maximum fluorescence level recorded in that stack (and their overlap is represented in the images as white pixels). The time-course of hypoxic modulation of $Na_V1.2$ current in moving from $O_2$ of 21% to 5% was studied with steps from $-100$ mV to $-20$ mV every 10 s and normalized to cell capacitance (pA/pF). Data represent 5–8 cells and biophysical parameters and single particles statistical analyses are summarized in *Supplementary file 1a* and *Table 2*, respectively. (**a**) Hypoxia rapidly recruits SUMO1 to the cell surface at sites with $Na_V1.2$ channels. The images show that the surface density of m-SUMO1 (top, left) is low compared to $CFP-Na_V1.2$ (top, middle) with little co-localization (top, right) in ambient $O_2$. After 40 s of hypoxia, rapid recruitment of SUMO1 (bottom, left) to the surface is observed to be at sites with $Na_V1.2$ channels (bottom, left). Surface levels of $Na_V1.2$ were not observed to change when cells were exposed to hypoxia (bottom, middle). (**b**) Histogram of surface density summarizing seven cells studied as described in **a** and *Table 2*. The density of pixels per $\mu m^2$ with SUMO1 alone (red) was $4 \pm 5$, with $Na_V1.2$ alone (blue) was $340 \pm 16$, and with both subunits was $67 \pm 6$ (red/blue hatch). Hypoxia increased co-localization to $268 \pm 12$ pixels per $\mu m^2$ and decreased the density of free $Na_V1.2$ channels ($139 \pm 8$) without altering the density of free SUMO1 ($12 \pm 2$). (**c**) The hypoxia-induced increase in the surface density of single fluorescent particles with both SUMO1 and $Na_V1.2$ was not observed in cells with 100 **n**M SENP1 in the pipette. (**d**) Hypoxia-induced increase in the surface density of SUMO1 was not observed in cells expressing CFP-tagged $Na_V1.2-Lys38Gln$ (K38Q). (**e**) The time-course for hypoxia-induced increase co-localization of $Na_V1.2$ and SUMO1 (Manders' coefficient, solid circle) and current-density (pA/pF, open circle) were coincident. The mean Manders' coefficient of $0.22 \pm 0.08$ measured in ambient $O_2$ increased to $0.72 \pm 0.12$ in less than 40 s of acute hypoxia. The current density increased from $-120 \pm 8$ to $198 \pm 10$ pA/pF. Increases in the Manders' coefficient and current density were unchanged 10 min after cells were restored to ambient $O_2$. (**f**) Hypoxia-induced increase in the surface density of colocalized SUMO1 and $Na_V1.2$ was also observed when cells were treated with 5 $\mu$m tetrodotoxin (TTX), a level that blocked over 95% of the $Na^+$ current.

the total to 65% ($265 \pm 12/\mu m^2$) with 35% unmodified by the tagged SUMO ($137 \pm 8/\mu m^2$). Although hypoxia increased the surface density of mCherry-SUMO1 associated with $CFP-Na_V1.2$ by ~4-fold, there was no change in the level of $CFP-Na_V1.2$ ($407/\mu m^2$ versus $402/\mu m^2$) and free mCherry-SUMO1 on the surface remained at low levels ($4/\mu m^2$ versus $12/\mu m^2$). During 20 min of post-hypoxia recovery in ambient $O_2$, the total number of $CFP-Na_V1.2$ surface particles decreased by ~10% without a change in the fraction of channels localized with mCherry-SUMO1 (*Table 2*). Because current

density decreased by just ~2% over the same period, we suspected that the decrease in fluorescent $Na_V1.2$ particles on the surface was due to CFP photobleaching, in-keeping with the idea that neither acute hypoxia-induced SUMOylation nor re-oxygenation led to net addition or removal of CFP-$Na_V1.2$ channels from the CHO cell plasma membrane; a conclusion supported as well by the similar size of CFP-$Na_V1.2$ currents recorded at 0 mV in ambient $O_2$ and 5% $O_2$ (*Supplementary file 1a*).

Corroborating a role for SUMOylation of $Na_V1.2$ in the response to hypoxia, 100 nm SENP1 suppressed recruitment of mCherry-SUMO1 to the cell surface ($3 \pm 2/ \mu m^2$), and abolished changes in peak current density, $V_{1/2}$ and SSI in response to hypoxia (*Figure 8c* and *Table 2*). Similarly, expression of CFP-$Na_V1.2$-Lys38Gln channels ($309 \pm 12/ \mu m^2$) eliminated recruitment of mCherry-SUMO1 to the cell surface ($4 \pm 2/ \mu m^2$) and showed no changes in biophysical parameters in response to hypoxia (*Figure 8d* and *Table 2*).

Consistent with a direct mechanism of action, the time course for the increase in $Na_V1.2$ channel current matched the rate of appearance of mCherry-SUMO1 at the plasma membrane with CFP-$Na_V1.2$ (*Figure 8e* and *Table 2*). Under ambient conditions, mCherry-SUMO1 and CFP-$Na_V1.2$ had a mean Manders' coefficient of co-localization of 0.22 and this increased to 0.72 over 40 s of hypoxia, showing a rate of rise of $17 \pm 3\%$ per 10 s. Similarly, channels formed with CFP-$Na_V1.2$ and mCherry-SUMO1 showed a time course for increased mean peak current-density of $15 \pm 2\%$ per 10 s (from $-120 \pm 8$ pA/pF to $-198 \pm 10$ pA/pF at $-20$ mV over 40 s, ~70% increase). Both the Manders' coefficient and current density were stable during 5 min of hypoxia and 10 min of subsequent recovery in ambient $O_2$.

Hypoxia-induced the SUMOylation of CFP-$Na_V1.2$ channels by co-expressed mCherry-SUMO1 to a steady state of ~65–80% of the channels in <1 min in CHO cells (*Figures 7* and *8*). That the entire population of surface CFP-$Na_V1.2$ channels was not SUMOylated despite ongoing hypoxia suggested that either ~20–35% of the channels were already modified on Lys38 by native, non-fluorescent SUMO in ambient $O_2$, and hence not reported by the assays, or these channels were resistant to SUMOylation if, like some other targets, SUMO modification of $Na_V1.2$ can be down-regulated by prior phosphorylation (*Konopacki et al., 2011*), ubiquitination, methylation, or acetylation (*Hendriks et al., 2014*). The rough concordance of the increase in current-density (~70%, *Table 2*) and in the fraction of newly SUMOylated channels in response to hypoxia (~50–60%) should be noted with some caution because the intracellular cesium fluoride solution used for recording $Na^+$ currents is very different from the native cytosolic milieu of intact cells studied with TIRF.

Finally, we confirmed that hypoxia-induced SUMOylation of CFP-$Na_V1.2$ in CHO cells was not triggered secondary to an increase in $Na^+$ flux through the channels by showing that the increase in surface levels mCherry-SUMO1 in response to acute hypoxia, and its co-localization with CFP-$Na_V1.2$, were insensitive to 5 μm TTX, a concentration that inhibited >95% of the $Na^+$ current (*Figure 8f*).

## Discussion

Here, we show that SUMOylation of $Na_V1.2$ channels at the plasma membrane underlies the immediate increase in $I_{Na}$ in response to acute hypoxia in cultured neurons and CHO cells. Previously, we demonstrated that SUMOylation increases excitability by suppressing the activity of background $K^+$ channels (*Rajan et al., 2005*; *Plant et al., 2010*, *2012*) and voltage-gated $K^+$ channels (*Plant et al., 2011*) that stabilize cells below firing threshold and restore cells to baseline after excitatory events, respectively. Now, we report that SUMOylation augments the activity of voltage-gated $Na_V$ channels that initiate and propagate action potentials, expanding the known targets through which the SUMO pathway can regulate neuronal activity. In addition, while chronic stressors, including low tissue oxygenation, have been associated with global changes in SUMOylation, this is the first report, to our knowledge, to demonstrate rapid modulation of the SUMO pathway over just 10's of s.

Hypoxia increases the SUMOylated fraction of $Na_V1.2$ channels at the cell surface in both CGN and CHO cells ~4-fold (*Figures 2* and *8*). Further, hypoxia or SUMO in the pipette shift the $V_{1/2}$ of channel activation and SSI leftward, increasing $Na^+$ current in response to depolarization by ~70% in both cell types (*Figures 1* and *5*, *Table 1*). Consistent with tonic regulation by the SUMO pathway of both CGN $I_{Na}$ and $Na_V1.2$ in CHO cells, $V_{1/2}$ in ambient $O_2$ rests midway between the SUMOylating and deSUMOylating experimental conditions; the full excursion in $V_{1/2}$ between the two conditions was ~27 mV in both cell types; and SENP1 decreased $Na^+$ current from baseline by ~75% in both

cases (*Figures 1* and *5*, *Table 1*). Also concordant are the rapid rates of hypoxia-induced increase of CGN $I_{Na}$ and $Na_V$1.2 current in CHO cells (*Figures 1* and *8*). Together, the following observations support the notion that hypoxia acts directly via SUMOylation of $Na_V$1.2 Lys38: modification of Lys38 is demonstrated directly by FRET and MS (*Figures 4* and *6*); hypoxia does not alter the current passed by $Na_V$1.2-Lys38Gln channels nor $I_{Na}$ in neurons pretreated with SUMO1 (*Figures 1* and *5*); hypoxia cannot overcome suppression of $I_{Na}$ by SENP1 (*Figures 1* and *5*); and, the time-course of SUMOylation of Lys38 matches the increase in $Na_V$1.2 currents observed when CHO cells are subjected to hypoxia (*Figure 8*). That hypoxia-mediated SUMOylation of $Na_V$1.2 is observed in the presence of TTX shows that $Na^+$ flux is not a prerequisite for channel SUMOylation. These findings offer a molecular rationale for acute, hypoxia-induced increases in the activity of neuronal $Na_V$ channels (*Boening et al., 1989*; *Horn and Waldrop, 2000*; *Raley-Susman et al., 2001*; *Hammarström and Gage, 2000*), a well-known trigger for the subsequent cascade of pathological events leading to cellular injury (*Banasiak et al., 2004*; *Fung and Haddad, 1997*).

Even short periods of hypoxia can diminish neuronal viability and perturb physiology. Thus, in perinatal rodents, brain hypoxia leads to cerebellar abnormalities, including a decrease in the number of Purkinje cells and interneurons and the thickness of the molecular and granule cell layers (*Biran et al., 2011*). Because inhibition of $I_{Na}$ by TTX reduces hypoxia-induced death in neuronal cultures (*Boening et al., 1989*; *Stys et al., 1992*; *Weber and Taylor, 1994*; *Xie et al., 1994*; *Taylor et al., 1995*; *Fung et al., 1999*; *Horn and Waldrop, 2000*; *Raley-Susman et al., 2001*; *Banasiak et al., 2004*), the response we study here — acute hypoxic recruitment of SUMO to $Na_V$1.2 channels leading to increased CGN $I_{Na}$— might be expected to be deleterious. However, neuronal pathology depends on the etiology, severity, and duration of the insult (*Puyal et al., 2013*). Thus, increased $I_{Na}$ may be self-limiting (or protective) if it leads to membrane depolarization, inducing $Na_V$ channel inactivation and, thereby, limits subsequent $Na^+$ flux. Consistent with this notion, we previously found that acute hypoxic challenge induces membrane depolarization when rat CGN were studied in current-clamp mode (*Plant et al., 2002*).

Indeed, chronic stressors that increase the level of SUMOylated proteins in cells (*Shao et al., 2004*), including the transcription factor HIF1α (*Cheng et al., 2007*), and modify global patterns of SUMOylation (*Cimarosti et al., 2012*) have been inferred to be cytoprotective in cases where gene regulation and protein turnover are altered to conserve energy (*Henley et al., 2014*). Thus, ground squirrel hibernation torpor, a model system for studying tissue protection during decreased blood flow and oxygen delivery, is associated with more SUMOylation in the brain, liver, and kidneys (*Lee et al., 2007*). Also consistent with a long-term protective role, cortical neurons isolated from mice and subjected to oxygen-glucose deprivation show decreased survival when the translation of SUMO2/3 is suppressed (*Datwyler et al., 2011*), and increasing SUMOylation by overexpression of Ubc9 is protective in a mouse model of focal cerebral ischemic damage (*Lee et al., 2011*). This report on acute regulation of $Na_V$1.2 by hypoxia joins an emerging body of work exploring the roles of SUMOylation in the nervous system in health and disease (*Henley et al., 2014*; *Cimarosti and Henley, 2008*).

Neuronal excitability is determined by the interplay of currents that depolarize and restore the membrane to resting potential. It is notable, therefore, that the immediate effects of ion channel SUMOylation we have observed thus far are excitatory. Here, we show that SUMOylation increases $I_{Na}$ in rat CGN, favoring depolarization and action potential firing. Our earlier work on $I_{Kso}$ in rat CGN carried by channels with K2P1 subunits showed SUMOylation to decrease that background $K^+$ conductance that stabilizes cells below firing threshold (*Plant et al., 2012*). So, too, we found SUMOylation to suppress the voltage-gated, delayed-rectifier $K^+$ current $I_{DR}$ in rat hippocampal neurons that speeds recovery after excitatory events, due to a rightward shift in the G-V relationship for activation of $K_V$2.1 channels (*Plant et al., 2011*). Of note, we found that SUMO did not regulate hippocampal pyramidal cell $I_{Na}$, a current carried predominantly $Na_V$1.1 and $Na_V$1.6 (*Gong et al., 1999*; *Vacher et al., 2008*). This suggests that SUMO regulation of $Na_V$ channels may be isoform-specific, a notion supported by the failure of the small amount of $I_{Na}$ in rat CGN that was not carried by $Na_V$1.2 to respond to hypoxia, SUMO1 or SENP1, a current we attribute primarily to $Na_V$1.6 based on its pharmacology.

Here, we observe that $Na_V$1.2 current remains elevated due to SUMOylation for at least 10–20 min following return to ambient $O_2$ in both CGN and CHO cells (*Figures 1* and *5*). In contrast, an acute hypoxic challenge to hypothalamic neurons induced an increase in $Na_V$ current that

recovered when the cells were returned to normoxia (*Horn and Waldrop, 2000*). While different targets have been reported to be SUMOylated in a stable (*Bernier-Villamor et al., 2002*) or labile (*Lang, 2015*) fashion, it is not necessarily the case that the response in hypothalamic neurons is mediated by the SUMO pathway and, thus, those findings may represent the operation of other Na$_V$ channel subtypes, differential effects of cellular history and environment, or variations in experimental strategies. Indeed, the cellular effects of hypoxia on neurons are numerous and those with significant influence should be expected to vary acutely and over time as well as by cell type.

Some of the limits of this study include that we evaluate CGN in vitro, isolated from their rich native environment and neuronal inputs. Furthermore, the kinetics of channel activity and SUMOylation-deSUMOylation we measure here at room temperature will be slower than the rates in vivo. Further, CGN cultures do not contain Purkinje cells or, as a result, the parallel fibers that arise from CGN to form synapses with Purkinje cells. Because parallel fibers are rich in Na$_V$1.2 channels, our studies may underestimate the effects of Na$_V$1.2-SUMOylation in response to hypoxia in the cerebellum. Moreover, we employ standard, potassium-free, recording solutions to study $I_{Na}$ and, therefore, $I_{Kso}$, a rat CGN current we previously showed to be responsive to hypoxia (*Plant et al., 2002*), was not evaluated in this work.

Despite extensive discovery efforts, medications offering broad neuroprotective efficacy in patients with brain hypoxia and ischemia are lacking (*Rogalewski et al., 2006*). The reasons underlying this therapeutic deficit reflect, in part, the complexity and variety of neurotoxic challenges. Our findings suggest that Na$_V$1.2 channels and SUMO pathway enzymes may be valuable therapeutic targets for reducing damage due to acute hypoxic insults to central neurons.

## Materials and methods

### Molecular biology and reagents

Rat Na$_V$1.2 (NM_012647.1) was handled in pcDNA1, as previously described (*Plant et al., 2006*). Human SUMO1$_{101}$ (BCOO5899) was amplified from a brain cDNA library (Clontech, Mountain View, CA) and inserted into pMAX as described before (*Rajan et al., 2005*). Sequences encoding mCherry or CFP were inserted as described (*Plant et al., 2010*) at the N-terminus of SUMO1 or Na$_V$1.2 respectively. Mutations were introduced with Pfu Quikchange PCR (Agilent, Santa Clara, CA). Purified SUMO1$_{97}$ and SENP1 were purchased from Boston Biochem (Cambridge, MA). SUMO1$_{97}$-T95K (introducing a diagnostic trypsin cleavage site) was produced as a His-tagged protein fused to the TEV cleavage domain in pET28a using the bacterial strain BL21 (DE3) and isolated by routine procedures cleaving the His-tag with TEV Protease before dialysis against 140 mm KCl, 0.5 mm CaCl$_2$, 5 mm HEPES, pH 7.4. Protein concentration was determined by BCA assay (Thermo Fisher, Waltham, MA). The sodium channel blockers, tetrodotoxin, 4–9-anhydrotetrodotoxin and μ-conotoxin-TIIIA were purchased from Tocris (Bristol, U.K.), Focus Biomolecules (Plymouth Meeting, PA) and CS Bio (Menlo Park, CA), respectively, and handled in buffers with 0.1% BSA.

### Cell culture and heterologous expression

CHO-K1 cells (RRID: CVCL_0214) were purchased from ATCC, identity-authenticated by cytochrome oxidase one analysis, demonstrated to be mycoplasma free by Hoeschst DNA stain and agar by culture, and maintained in F12K medium supplemented with 10% FBS (ATCC, Manassas, VA). Plasmids were transfected into cells with Lipofectamine 2000 according to the manufacturer's instructions (Thermo Fisher). Experiments were performed 36 to 48 hr post transfection at room temperature. The rat $\beta$1 accessory subunit was co-expressed with Na$_V$1.2 subunits unless otherwise indicated. CGN were cultured from 6 to 8 day old Sprague-Dawley rat pups (Charles River, Wilmington, MA; RRID:RGD_734476) as previously described (*Plant et al., 2002*). Briefly, the cerebellum was isolated, cut into ~300 μm cubes, triturated with a fire-polished Pasteur pipette and incubated for 15 min at 37°C with 2.5 mg/ml trypsin in PBS. Digestion was halted by the addition of PBS containing 0.1 mg/ml soybean trypsin inhibitor, 2000 U/ml DNase I and 1 mm MgCl$_2$. Cells were pelleted by centrifugation for 1 min at 100 $g$, resuspended in MEM supplemented with 10% FBS, 26 mm glucose, 19 mm KCl and 2 mm L-glutamine then seeded on poly-L-lysine–coated coverslips and incubated at 37°C in a humidified atmosphere containing 95% air (21% O$_2$) and 5% CO$_2$, or in a tri-gas incubator (Heracell, Thermo Fisher) set to 7% O$_2$ and 5% CO$_2$, as indicated. To minimize reoxygenation prior to the

experiment, cells incubated at 7% $O_2$ were transferred to the patch-clamp microscope in a sealed container equilibrated to 7% $O_2$. After 48 hr, the medium was exchanged for MEM supplemented with 10% horse serum, 26 mm glucose, 19 mm KCl, 2 mm L-glutamine and 80 μm L-fluorodeoxyuridine to prevent the proliferation of non-neuronal cells. CGN were studied between days 7 and 10 in culture.

## Electrophysiology

$I_{Na}$ and whole-cell $Na_V1.2$ channel currents were recorded using an Axopatch 200B amplifier and pCLAMP software (Molecular Devices, Sunnyvale, CA) at filter and sampling frequencies of 10 and 50 kHz respectively. Cultured CGN or CHO cells were superfused with a solution comprising (in mm): NaCl 130, CsCl 4, $CaCl_2$ 2, $MgCl_2$ 1.2, glucose 5.5, HEPES 10 and, 100 μM $CdCl_2$, a concentration previously shown to block >98% of voltage-gated calcium channel current in CGN (*Pearson et al., 1993*). The pH was adjusted to 7.4 with NaOH and HCl. Cells were studied at room temperature with borosilicate glass pipettes (Warner, Hamden, CT) with a resistance of 2–3 MΩ when filled with a solution comprising (in mm): CsCl 60, CsF 80, $CaCl_2$ 1, $MgCl_2$ 1, $Na_2ATP$ 5, EGTA 10, HEPES 10, pH 7.4 with CsOH. Pipettes were coated with Sylgard (Sigma-Aldrich, St. Louis, MI) prior to use. Capacitance artifacts were subtracted online, series resistance was compensated to 70% and cells with a series resistance of less than 10 MΩ were studied. Current-voltage relationships were evoked from a holding potential of −100 mV by 30 ms test pulses between −75 and 20 mV, in 5 mV increments. Steady-state inactivation was studied by holding cells at −100 mV and then comparing currents evoked by 30 ms test pulses between −110 mV and 0 mV to those evoked by a 50 ms prepulse to 0 ms. A 10-s interpulse interval was used in both cases. Normalized peak current values are plotted against prepulse potential (mV). A Boltzmann function, $I = I_{max}/(1 + exp[V – V_{½}/k])$, where $I_{max}$ is the maximum current and $k$ is slope factor, was used to fit normalized activation-voltage relationships. Recovery from fast inactivation was studied by holding cells at −100 mV and comparing currents evoked by a pair of 30 ms test step to −20 mV separated by an interpulse interval that increased in duration by 30 ms increments per sweep. The time constant for recovery from inactivation ($\tau_{Recovery}$) was obtained from mono-exponential fits of the normalized current amplitude to the recovery time using $I = I_0 + A(_{exp}^{(t/\tau Recovery)})$, where A is the amplitude of components and $t$ is time. Whole-cell currents were normalized to cell capacitance. Mean ± S.E.M. capacitance values were 6 ± 2 pF for CGN and 11.5 ± 2 pF for CHO cells.

## Acute hypoxia for electrophysiology

Acute hypoxia was achieved as described previously (*Plant et al., 2002*). Briefly, the cells were made hypoxic (5% or 1.5% $O_2$ as indicated) by switching the perfusate with one that had been bubbled with nitrogen for at least 30 min prior to perfusion. Oxygen tension was measured at the cell by a polarized carbon fiber electrode; solution exchange occurred in less than 10 s.

## Two-color TIRFM

Single protein complexes at the surface of live CHO cells were identified using TIRFM as previously described (*Plant et al., 2014*). Cells were studied in a solution comprising (in mm): NaCl 130, KCl 4, $MgCl_2$ 1.2, $CaCl_2$ 2, HEPES 10, pH was adjusted to 7.4 with NaOH. The critical angle for TIRF was adjusted using a CellTIRF illuminator (Olympus, Waltham, MA) and a high numerical aperture apochromat objective (150x, 1.45 NA) mounted on an automated fluorescence microscope controlled by Metamorph software (Molecular Devices). For simultaneous illumination of two fluorophores, Cell-TIRF software (Olympus) was used to adjust the critical angle for each excitation wavelength to generate evanescent waves of equivalent depth (100 nm). mCherry was excited by a 561 nm laser line and CFP was excited a 445 nm laser line. When CFP was studied with mCherry the emitted light signals were split using a 520 nm dichroic mirror mounted in a DualView adapter (Photometrics, Tucson, AZ), and each wavelength was directed to one-half of an EM-CCD. The dichroic mirror was disengaged when single fluorophores were studied.

To assess stoichiometry, fluorophores were photobleached by continual excitation and data were captured as movies of 100–400 frames acquired at 1 Hz. When CFP was studied with mCherry in the same cell, the data for each fluorophore were saved as separate stacks and processed in an identical

manner. Images were background corrected by subtracting the mean of five fully bleached frames from the end of each stack analyzed. Misalignment of the data between stacks was corrected in ImageJ using StackReg. Fluorescent spots were defined as a discrete 3 × 3 pixel region around a pixel of maximum intensity, as before (*Plant et al., 2010*, *2014*). Fluorescence is reported as the change in fluorescence intensity normalized by the initial fluorescence for each trace. The density of co-localized and single fluor-fluorescent particles was determined following thresholding and watershed separation in ImageJ. The particle number was counted in 4, separate 100 by 100 pixel regions of interest for 6–10 cells per group using the Analyze particles plugin.

Manders' coefficient of colocalization was assessed from live-cell simultaneous two-color TIRF images captured at 5 s intervals to minimize photobleaching. Data stacks were background subtracted and aligned for each fluorophore post-hoc, as above. Co-localization of partner pixels from the two stacks of images was defined as the presence of both fluorophores with at least 30% of maximum fluorescence levels recorded in that region of interest. Mean Manders' coefficients were calculated for 3–5 separate 100 by 100 pixel regions of interest per cell.

## amFRET immunocytochemistry and microscopy

Double immunostaining for SUMO1 (RRID:AB_2198257) and Na$_V$1.2 (RRID:AB_10673401), or the housekeeping gene, GAD67 (RRID:AB_2278725) was performed to allow measurement of FRET between secondary antibodies labeled with Alexa 488 and Alexa 594, employed as FRET donor and acceptor fluorophores, respectively (*Plant et al., 2011*). CGN were permeabilized with digitonin (20 mg/ml) in a HEPES-buffered solution matching the intracellular ionic composition containing EGTA (1 mm) and a cocktail of protease inhibitors (N-methylmaleimide, PMSF, aprotinin, leupeptin, antipain, and pepstatin A), and then fixed in 2% paraformaldehyde for 30 min. SUMO1 or GAD67 were detected with highly cross-adsorbed donkey anti-rabbit IgG (1 µg/ml, Thermo Fisher), labeled with Alexa 594 to act as the FRET acceptor. Na$_V$1.2 antibody binding was detected with highly cross-adsorbed donkey anti-goat IgG (1 µg/ml, Thermo Fisher) labeled with Alexa 488 to act as a FRET donor. Neurons were exposed to hypoxia by overnight equilibration of dishes of HCO$_3$-buffered saline matching the osmolality and ionic composition of the culture media in an environmental chamber (Coy Laboratory Products, Grass Lake, MI) with an atmosphere of 1% O$_2$ and 5% CO$_2$ at 37°C. Neurons on coverslips were immersed in the hypoxic buffer for 5 min and were then permeabilized and fixed in similarly equilibrated HCO$_3$-buffered intracellular saline containing digitonin.

For amFRET microscopy, neurons were imaged with a 60x, NA 1.40 oil objective on a Leica SP5 laser scanning confocal microscope with identical illumination acquisition settings across staining conditions. Sequential confocal images of Alexa 488 and Alexa 594 were obtained using laser lines at 488 nm and 546 nm, respectively. The Leica Acceptor Photobleaching Wizard was used to obtain pre-bleach images of donor (Na$_V$1.2, Alexa 488) and acceptor (SUMO1 or GAD67, Alexa 594) intensities to create a region to bleach on the donor subunit image; to bleach the Alexa 594 fluorescence within the region (200 passes of full laser power); and to obtain post-bleaching images of Alexa 488 and Alexa 594 fluorescence. Image pairs demonstrating a focus shift between pre- and post-bleaching images were excluded. Twelve-bit raw confocal images were deconvolved using Huygens Deconvolution software (Scientific Volume Imaging, Hilversum, Netherlands) and Maximum Likelihood Estimation with a signal-to-noise ratio of 20. Using ImageJ, raw Alexa 488 images were de-noised, and FRET efficiency images within the bleaching regions created, on a voxel-by-voxel basis, by subtracting pre-bleach from post-bleach and then dividing the difference by the post-bleach intensity.

## STORM immunostaining and imaging

Cells were first double-immunostained for SUMO1 and Na$_V$1.2 using the same antibodies as above. Secondary antibodies against anti-SUMO1 and anti-Na$_V$1.2 were labeled here with Alexa 568 and Alexa 647, respectively. For permeabilization, digitonin was replaced with saponin (0.01%), which was present in all buffers during staining. To label the plasma membrane, cells were incubated in saponin-free buffer for 2 hr to allow the lipid membrane to regain its integrity and then incubated with biotin (2 mM) for 30 min, followed by Alexa 488-labeled streptavidin (5 µg/ml; Thermo Fisher) for 1 hr all at room temperature.

For microscopy, coverslips were mounted on slides in PBS containing cysteamine (MEA, 100 µM) and sealed with epoxy. Neurons were imaged on the stage of a Leica SR GSD 3D inverted

microscope with a 160x, 1.43 NA objective. Fluorophores were sequentially excited, from red to blue, employing 642, 532, and 488 nm lasers, respectively, and individual fluorescent events imaged with an Andor iXon Ultra EMCCD over a 180 × 180 pixel region. For each fluorophore, images were collected at 56 Hz for at least 10 min. Image stacks were background subtracted and Gaussian blurred in FIJI (*Schindelin et al., 2012*), and reconstructed with the Thunderstorm Plug-in (*Ovesný et al., 2014*) to produce a list of fluorescent molecule localizations. Fluorescent events having X-Y uncertainty >40 nm or a Z uncertainty >400 nm were filtered out, and a 20-image stack of X-Y scatter plots to encompass a volume of 41 × 41 × 800 nm were produced for each fluorophore. Maximum Z-projections for each fluorophore were produced and overlaid and masks consisting of fluorescent events from all three fluorophores occurring in the same voxel were produced.

## Live CHO cell FRET microscopy

Donor-decay time-course was studied as before (*Plant et al., 2010*), using an automated Olympus IX81 epi-fluorescence microscope. Cells were studied in a solution comprising (in mM): NaCl 130, KCl 4, $MgCl_2$ 1.2, $CaCl_2$ 2, HEPES 10, pH was adjusted to 7.4 with NaOH. CFP was excited at 458 nm and the emission collected through a 470–500 nm bandpass filter, YFP was excited at 514 nm and the emission collected through a 525–575 nm filter. Images were captured using a CCD camera controlled by Metamorph (Molecular Devices) and were analyzed with ImageJ.

## Mass spectrometry

Rat $Na_V1.2$ N-terminal residues 1–125 were cloned into pET28a vector with six-His residues and a tobacco etch virus (TEV) cleavage site replacing the thrombin site and expressed in BL21(DE3) *E. Coli* with a vector carrying mouse E1 (as a linear fusion product of Aos1 and Uba2), E2 (Ubc-9) and $SUMO1_{1-97}$ with a T95K mutation so that trypsin digestion before MS leaves a Gly-Gly tag on the ε-amino group of target lysines as before (*Plant et al., 2010*). Protein was purified with Ni-NTA affinity columns to a yield of ~1 mg of $Na_V1.2_{1-336}$–$SUMO1_{1-97}$T95K per 1 L media. In-gel protein trypsin digestion was performed after SDS-PAGE. Samples were analyzed with a Dionex Ultimate 3000 Nano-HPLC System (Sunnyvale, CA), Zorbax C18 trapping and analytical columns, and an LTQ-FT tandem MS instrument (Thermo Fisher) equipped with a nanospray source and a picotip nanospray needle (8 μm id tip; New Objectives, Woburn, MA) running Xcalibur software. Spectra were acquired using positive ion nano ESI mode with the FT-ICR acquiring precursor spectra from 250–1800 m/z at a resolution of 50000 at m/z 400. Tandem mass spectra were acquired in a data-dependent manner using the five most intense ions with charge states of +2 or higher from each FT-ICR MS scan to trigger the LTQ performance of collision-induced dissociation (CID) on each of the selected precursor ions using an activation Q of 0.25, a normalized collision energy of 35, and an activation time of 30 ms. The RAW data files from each run were processed with DTA supercharge to generate MGF peak list files and ReadW to produce binary mzXML files (http://tools.proteomecenter.org/software.php). MGF files were submitted to a Mascot sequence database (Matrix Science, Boston, MA) and mzXML files to the Sagen Research Sorcerer database. Searches were then run against a custom database created by adding the sequences of rat $Na_V1.2$ and $SUMO1_{1-97}$T95K to the International Protein Index (IPI) - Human database version 3.33 (www.ebi.ac.uk). In each search, a peptide precursor mass tolerance of 5 ppm was used, allowing for modifications due to additions such as Gly-Gly (+114.04292, Th), up to three missed cleavages, and strict adherence to tryptic digestion rules. Search results were then loaded into Scaffold software (Proteome Software, Portland, OR) and peptides with scores of 95% confidence or better used to confirm peptide assignments. The LTQ was programmed to accumulate precursor masses of expected peptides and perform dedicated tandem MS of only those ions using the LC conditions above and a multi-event study consisting of single MS scans on the FT-ICR and up to three dedicated MS/MS experiments on the selected masses. Data obtained were extracted as above and subjected to searches and manual evaluation including assessment of molecular weight, spectral quality and consistency of peptides retention time in relation to unmodified versions of the peptide.

## Data and statistical analysis

Data were analyzed using pClamp, Origin and Excel software. Data were assessed for statistical differences between groups by one-way analysis of variance with Bonferroni post hoc analysis to test differences within pairs of group means for all data set with an F-value of $p<0.05$.

## Acknowledgements

The authors thank A Arena, E Dowdell, D Araki, I Dementieva and J Wang for technical support. We are grateful to AL Goldin (UCI) for rat $Na_V1.2$ and RY Tsien (UCSD) for the mCherry clones. The work was funded by National Institutes of Health grants R01NS058505 to SANG and R01NS056313 to JDM.

## Additional information

### Funding

| Funder | Grant reference number | Author |
| --- | --- | --- |
| National Institute of Neurological Disorders and Stroke | R01NS058505 | Steven AN Goldstein |
| National Institute of Neurological Disorders and Stroke | R01NS056313 | Jeremy D Marks |

The funders had no role in study design, data collection and interpretation, or the decision to submit the work for publication.

### Author ORCIDs

Leigh D Plant, http://orcid.org/0000-0002-1622-1655
Jeremy D Marks, http://orcid.org/0000-0002-2644-5257
Steve AN Goldstein, http://orcid.org/0000-0001-5207-5061

### Ethics

Animal experimentation: University of Chicago Institutional Animal Care and Use Committee (IACUC) approved the use of vertebrate animals (rats) in protocol #68001 to J. Marks. Brandeis University Institutional Animal Care and Use Committee (IACUC), operating under Animal Welfare Assurance #A3445-01, approved the use of vertebrate animals (rats) in protocol #0910-09 to Suzanne Paradis. The neurons are studied by various techniques, including electrophysiology, immunocytochemistry and microscopy. Invertebrate model systems are not useful here because the proteins we study are not present in their genome. Where possible we use tissue culture cells, however, much of the work focuses on how native ion channels are expressed and regulated. Rodents are the lowest phylogenetic order in which we can carry out the experiments we propose. Rats are also the standard organism for studies of cerebellar granule neuron physiology and are well established in the field as the organism of choice for the studies proposed. Thus, there is an extensive literature against which to compare and interpret the experimental results. Rats are frequently the source of neurons for cultures due to the large size of their brains and the relative robustness of their neurons in culture. The rat is a well-accepted model for studying ischemia in vitro, providing several advantages: a) the vulnerability of the rodent nervous system to hypoxia-ischemia is well-characterized, b) that rat shares identical mechanisms of ischemia-induced neuronal death with humans and c) procedures for minimizing discomfort, distress, pain, and injury as well as for euthanasia are extensively studied and ours follow AVMA guidelines.

### Author contributions

LDP, Conceptualization, Methodology, Investigation, Formal Analysis, Writing – Original Draft, Writing – Review and Editing, Visualization; JDM, Methodology, Investigation, Formal Analysis, Writing – Review and Editing, Visualization, Funding Acquisition; SANG, Conceptualization, Methodology, Writing – Original Draft, Writing – Review and Editing, Visualization, Funding Acquisition, Supervision

## Additional files

### Supplementary files

• Supplementary file 1. The biophysical properties of $I_{Na}$ and $Na_V1.2$ channels. (a) Biophysical properties of $I_{Na}$ and $Na_V1.2$ channels cultured at 21% $O_2$. (b) Biophysical properties of $I_{Na}$ cultured at 7% $O_2$. (c) Biophysical properties of $I_{Na}$ and $Na_V1.2$ channels expressed without the $\beta$1-subunit.

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
