## [Decision Letter]

Thank you for submitting your article "SUMOylation of Na_V_1.2 channels mediates the early response to acute hypoxia in central neurons" for consideration by *eLife*. Your article has been favorably evaluated by Richard Aldrich as the Senior Editor and three reviewers, one of whom, Baron Chanda (Reviewer #1), is a member of our Board of Reviewing Editors.

The reviewers have discussed the reviews with one another and the Reviewing Editor has drafted this decision to help you prepare a revised submission.

Summary:

The mechanism for increased sodium leak, which is a major component of the initial response to hypoxia in the central nervous system, is not known. In this study, Plant et al. show that SUMOylation of the Na_V_1.2 channels within few tens of seconds of hypoxia underlies this excitatory shift in sodium conductance properties. They also provide direct evidence that SUMO1 interacts with Na_V_1.2 channel and that SUMOylation of a specific lysine in the N-terminus of these channels mediates these responses. Furthermore, they provide evidence that SUMO regulation of sodium current is isoform specific and Na_V_1.6 which is also found in rat CGN is not modulated by SUMO1. Overall, this is a solid contribution to the field and shows for the first time that the SUMO pathway is involved in rapid modulation of sodium currents in response to hypoxia.

Essential revisions:

The three reviewers found the study to be of great interest and well executed but were concerned about the overly broad generalizations especially in absence of animal studies. For instance, one of the reviewer notes that the bulk of Na_V_1.2s in CGNs in situ are in parallel fibers and terminals, and not in the cell bodies as is in the population studied by the authors in cultured CGNs. The authors do not study these neurons in situ and it is possible that the other SUMO regulated channels (K2P1, K_V_2.1) also play an important role in defining CGN excitability in culture and in situ. The reviewer also notes that Plant et al. (Stroke 33 (2002) 2324-8) defined TASK-1 (K2P3.1/KCNK3) as the only ion channel important in regulating the hypoxic depolarization of CGNs. In light of these comments, it is essential that authors provide a more nuanced discussion in broader context of the limitations of the present study.

In addition, the other essential points that should be addressed experimentally are:

1) The authors report peak sodium currents and normalized conductance-voltage curves, but they do not report the maximum sodium conductance for any of their experiments. They measure sodium current at -20 mV, which is about 50% activation for the control condition but manipulation of SUMO shifts the voltage dependence of activation so the changes in sodium currents measured at this fixed potential may be a mixture of changes in V_a_ and in G_max_. To me, it would be nice to see both parameters separately, especially since G_max_ changes could reflect changes in channel number on the cell surface, in contrast to their conclusions about regulation of channel activity.

2) In Figure 2 and Figure 4, I don't find the evidence for cell surface localization of the labeled channels very convincing. Most of the intracellular volume of these cells seems to be taken up by their nuclei, and it is not clear whether the fluorescent label is at the cell surface or in the cytosol between the nucleus and the cell surface.

3) Figure 2: it is not clear under what conditions (normoxic, hypoxic) the amFRET experiments were performed. The context of these n=1 results to the author's model for the signaling pathway is unclear, especially if this was done under normoxic conditions. A more comprehensive amFRET experiment comparing Na_V_1.2:SUMO1 amFRET levels quantitatively, under the conditions used in the experiments in Figure 1 that lead to changes in Na_V_1.2-based *I_Na_*(control, hypoxia, SUMO1, SENP1) would represent a more substantial contribution to this study.

4) The SDS gel in Figure 6 is not very convincing. The banding pattern is very complex (10-11 bands between the 25-50 kD range shown) and there is no apparent band at the site of the arrow, and the band at the asterisk is very minor. For a protein expressed in bacteria, cleaner and more robust proteins bands are expected. Adding a control lane showing uninduced cultures do not have the protein of interest will also help clarify this issue.

Details of additional points raised by the reviewers that can be clarified in the text are:

1) The last sentence of the Abstract gives the impression that hypoxia induced SUMOylation shifts the half-activation of the curve by -26.5 mV. This is not strictly true because compared to the half-activation of the wild type channels under ambient oxygen conditions the shift is about 9 mV in the hyperpolarizing direction.

2) In Figure 1, the authors show that the SENP1 deSUMOylase causes the conductance voltage curves to shift to the right compared to the wild type channels under ambient conditions. Does that imply that a significant fraction of the wild type channels are SUMOylated under normal oxygen conditions and that the observed current responses are ensemble averages of SUMOylated and unSUMOylated channels? While this notion is broadly supported by FRET data and single molecule photobleaching experiments, the numbers do not match. For instance, Figure 6 shows more than 3 fold increase in SUMO associated Na_V_1.2 under hypoxic conditions but the shift in G-V curves is less than 50%. Is it possible that some of this is due to expression in a heterologous system?

3) The results in Figure 3 are a bit confusing. The long-term culture in 7% O_2_ yields a larger shift in the activation V_1/2_ (-42.9 mV) that does acute 7% O_2_ treatment (-34 mV), yet does not lead to an increase on *I_peak_* measured at -20 mV. This does not conform to expectations of their model (left shifted G-V curves lead to enhanced INa). The manner in which these results are described in the Results subsection “CGN cultured at 7% O_2_” glosses over these findings, and suggests that going from 7% to 1.5% O_2_ is pretty much the same as going from 21% to 7%. This is not the case, and this point is critical to their study, as the data support that long term culture in 7% O_2_ already leads to substantial (SUMO-based?) changes in Na_V_1.2. This needs to be addressed.

4) Figure 5 legend: the statement "Na_V_1.2-Lys38Gln channels passed large currents" is a bit misleading, as the scale bars are different than in 5A, and a major point of this finding is that the currents are much smaller than WT Na_V_1.2. That the Na_V_1.2-Lys38Gln currents are in fact quite small should be more clearly stated in the legend.

---

## [Author Response]

*Essential revisions:*

*The three reviewers found the study to be of great interest and well executed but were concerned about the overly broad generalizations especially in absence of animal studies. For instance, one of the reviewer notes that the bulk of Na_V_1.2s in CGNs in situ are in parallel fibers and terminals, and not in the cell bodies as is in the population studied by the authors in cultured CGNs. The authors do not study these neurons in situ and it is possible that the other SUMO regulated channels (K2P1, K_V_2.1) also play an important role in defining CGN excitability in culture and in situ. The reviewer also notes that Plant et al. (Stroke 33 (2002) 2324-8) defined TASK-1 (K2P3.1/KCNK3) as the only ion channel important in regulating the hypoxic depolarization of CGNs. In light of these comments, it is essential that authors provide a more nuanced discussion in broader context of the limitations of the present study.*

Thank you. The Discussion has been made more complete by the addition of a sentence in the second paragraph and two paragraphs (sixth and seventh) that consider, in part, other channels known to respond to hypoxia in CGN, other neuronal responses to hypoxia, and four important study limitations including the absence of Purkinje cells in our culture and, as a result, the absence of parallel fibers that carry many Na_V_1.2 channels in cerebellum in vivo.

*In addition, the other essential points that should be addressed experimentally are:*

*1) The authors report peak sodium currents and normalized conductance-voltage curves, but they do not report the maximum sodium conductance for any of their experiments. They measure sodium current at -20 mV, which is about 50% activation for the control condition but manipulation of SUMO shifts the voltage dependence of activation so the changes in sodium currents measured at this fixed potential may be a mixture of changes in V_a_ and in G_max_. To me, it would be nice to see both parameters separately, especially since G_max_ changes could reflect changes in channel number on the cell surface, in contrast to their conclusions about regulation of channel activity.*

Thank you, this feedback led to inclusion of important data and clarifying text. First, we amend Table 1 to include comparisons at 0 mV for CGN *I_Na_* and for Na_V_1.2 expressed in CHO cells; at this voltage, the G-V relationships are saturated for all test conditions and the currents are ~ equal supporting the hypothesis that changes in the G-V relationships are sufficient to rationalize the rapid changes in current (that is, without a need to invoke changes in channel number or unitary current). In the legend, we also give the voltage for peak current and values for CGN cultured at 21% O_2_ with all the pipette solutions (i.e., control, SUMO1 and SENP1). We also include comparisons at 0 mV for CFP-Na_V_1.2 in CHO cells in [Supplementary-material SD1-data]. The new data are described in the subsection “The SUMO pathway regulates CGN *I_Na_*”, last paragraph, a revised sentence in the subsection “SUMOylation of Na_V_1.2 on Lys38 is necessary and sufficient for the hypoxic response” (on Na_V_1.2 in CHO cells), and a revised sentence in the subsection “Hypoxia induces rapid monoSUMOylation of Na_V_1.2 channels” (on CFP-Na_V_1.2 in CHO cells where TIRF also supports the conclusion that rapid trafficking is not a primary response to acute hypoxia since the surface density of channels did not change over 5 min).

*2) In Figure 2 and Figure 4, I don't find the evidence for cell surface localization of the labeled channels very convincing. Most of the intracellular volume of these cells seems to be taken up by their nuclei, and it is not clear whether the fluorescent label is at the cell surface or in the cytosol between the nucleus and the cell surface.*

Thank you; this feedback led to revision of Figure 4 as well as inclusion of new data employing three-color GSD-STORM super resolution microscopy (SRM) in Figure 2. First, Figure 4 is reformatted to improve clarity of the image even when it is this small so the CHO cell plasma membrane is more readily seen. Second, new Figure 2 shows proximity of the CGN plasma membrane, Na_V_1.2 and SUMO1 with an X-Y range of < 20 nm using SRM. This new work is described in the Results subsection “Interaction of native Na_V_1.2 and SUMO1 at the CGN surface increases with hypoxia” and Methods subsection” STORM immunostaining and imaging”.

*3) Figure 2: it is not clear under what conditions (normoxic, hypoxic) the amFRET experiments were performed. The context of these n=1 results to the author's model for the signaling pathway is unclear, especially if this was done under normoxic conditions. A more comprehensive amFRET experiment comparing Na_V_1.2:SUMO1 amFRET levels quantitatively, under the conditions used in the experiments in Figure 1 that lead to changes in Na_V_1.2-based I_Na_ (control, hypoxia, SUMO1, SENP1) would represent a more substantial contribution to this study.*

Thank you; the feedback led to new amFRET studies comparing neurons under normoxic and hypoxic conditions that we now include. We sought originally to confirm that the two native proteins were found together at the neuron surface consistent with the electrophysiology so we stained CGN in ambient conditions. Now, new Figure 2 compares amFRET under normoxic conditions (showing basal levels of SUMOylation) and amFRET with hypoxia showing a 4-fold increase in SUMO with the channels; this, and the statistics underlying the studies (now including the number of neurons studied, n = 8-10), are described in the Results subsection “Interaction of native Na_V_1.2 and SUMO1 at the CGN surface increases with hypoxia”, the Methods subsection “amFRET immunocytochemistry and microscopy**”**, the legend for Figure 2 and Figure 2—figure supplement 1. To your last point, the new amFRET microscopy studies were performed under ambient and hypoxic conditions with native intracellular composition prior to fixation whereas in Figure 1 we replaced the cytosol with cesium fluoride (CsF) solution to allow patch clamp Na^+^ current recording; this limits direct comparison, a challenge noted again below and now recognized in the subsection “Hypoxia induces rapid monoSUMOylation of Na_V_1.2 channels”. (Note, to add the new data to Figure 2, toxin-blocking studies were moved from Figure 2 to Figure 1.)

*4) The SDS gel in Figure 6 is not very convincing. The banding pattern is very complex (10-11 bands between the 25-50 kD range shown) and there is no apparent band at the site of the arrow, and the band at the asterisk is very minor. For a protein expressed in bacteria, cleaner and more robust proteins bands are expected. Adding a control lane showing uninduced cultures do not have the protein of interest will also help clarify this issue.*

Thank you; this feedback led to inclusion of more gel lanes, correction of the main text and movement of the arrow and asterisk to the correct locations. Revised Figure 6 now compares expression of channel alone, SUMO alone, and channel with SUMO. The position of the arrow and the asterisk are shifted to highlight the bands correctly. Also, text is added to the legend to note and cite the complex band patterns seen in this system due to His-labeling of target and native proteins by expression of the full array of SUMO pathway enzymes and His-tagged SUMO).

*Details of additional points raised by the reviewers that can be clarified in the text are:*

*1) The last sentence of the Abstract gives the impression that hypoxia induced SUMOylation shifts the half-activation of the curve by -26.5 mV. This is not strictly true because compared to the half-activation of the wild type channels under ambient oxygen conditions the shift is about 9 mV in the hyperpolarizing direction.*

Thank you; we have correctly reworded the Abstract.

2) In Figure 1, the authors show that the SENP1 deSUMOylase causes the conductance voltage curves to shift to the right compared to the wild type channels under ambient conditions. Does that imply that a significant fraction of the wild type channels are SUMOylated under normal oxygen conditions and that the observed current responses are ensemble averages of SUMOylated and unSUMOylated channels? While this notion is broadly supported by FRET data and single molecule photobleaching experiments, the numbers do not match. For instance, Figure 6 shows more than 3 fold increase in SUMO associated Na_V_1.2 under hypoxic conditions but the shift in G-V curves is less than 50%. Is it possible that some of this is due to expression in a heterologous system?

Thank you; it is very important that we make clear that some channels are SUMOylated in ambient O_2_, and that hypoxia increases that fraction. We now do this early in the Introduction, in a dedicated new paragraph in the Results subsection “The SUMO pathway regulates CGN Ina” for CGN, later in the report in the section on CGN spectroscopy, subsection “Interaction of native Na_V_1.2 and SUMO1 at the CGN surface increases with hypoxia”, and again thereafter in the reconstituted CHO cell system (where it had previously been noted). The failure of numbers to match in the original has been addressed first by making it clear that we see close correlations between the native and reconstituted systems (as summarized in the subsection “Hypoxia induces rapid monoSUMOylation of Na_V_1.2 channels”) while emphasizing study limitations in the Discussion (seventh paragraph). You highlight a good example where the comparison was a challenge (now noted in the subsection “Hypoxia induces rapid monoSUMOylation of Na_V_1.2 channels”): current measured in CHO cells and CGN involves replacing the cytosol with cesium fluoride (CsF) solution for recording whereas CHO cells are intact for TIRF and therefore enjoy a native intracellular milieu (indeed, we found CsF recording solution precluded fluorescent measurements).

*3) The results in Figure 3 are a bit confusing. The long-term culture in 7% O_2_ yields a larger shift in the activation V_1/2_ (-42.9 mV) that does acute 7% O_2_ treatment (-34 mV), yet does not lead to an increase on I_peak_ measured at -20 mV. This does not conform to expectations of their model (left shifted G-V curves lead to enhanced INa). The manner in which these results are described in the Results subsection “CGN cultured at 7% O_2_” glosses over these findings, and suggests that going from 7% to 1.5% O_2_ is pretty much the same as going from 21% to 7%. This is not the case, and this point is critical to their study, as the data support that long term culture in 7% O_2_ already leads to substantial (SUMO-based?) changes in Na_V_1.2. This needs to be addressed.*

Thank you again for helping us to clarify. We edited the text in the subsection “CGN cultured at 7% O_2_” to make clear that 7-10 days at 7% changed the baseline attributes of the cells compared to those incubated at 21% but that subsequent changes with hypoxic challenge are similar in type and magnitude while starting from this new steady-state.

*4) Figure 5 legend: the statement "Na_V_1.2-Lys38Gln channels passed large currents" is a bit misleading, as the scale bars are different than in 5A, and a major point of this finding is that the currents are much smaller than WT Na_V_1.2. That the Na_V_1.2-Lys38Gln currents are in fact quite small should be more clearly stated in the legend.*

Thank you; the error is now corrected (large changed to small).